# Microscale cavitation as a mechanism for nucleating earthquakes at the base of the seismogenic zone

Berend A. Verberne [1], Jianye Chen [1], André R. Niemeijer [1], Johannes H.P. de Bresser[1], Gillian M. Pennock [1], Martyn R. Drury[1] & Christopher J. Spiers[1]

Major earthquakes frequently nucleate near the base of the seismogenic zone, close to the brittle-ductile transition. Fault zone rupture at greater depths is inhibited by ductile flow of rock. However, the microphysical mechanisms responsible for the transition from ductile flow to seismogenic brittle/frictional behaviour at shallower depths remain unclear. Here we show that the flow-to-friction transition in experimentally simulated calcite faults is characterized by a transition from dislocation and diffusion creep to dilatant deformation, involving incompletely accommodated grain boundary sliding. With increasing shear rate or decreasing temperature, dislocation and diffusion creep become too slow to accommodate the imposed shear strain rate, leading to intergranular cavitation, weakening, strain localization, and a switch from stable flow to runaway fault rupture. The observed shear instability, triggered by the onset of microscale cavitation, provides a key mechanism for bringing about the brittle-ductile transition and for nucleating earthquakes at the base of the seismogenic zone.

[1] Department of Earth Sciences, Utrecht University, Budapestlaan 4, 3584 CD Utrecht, The Netherlands. Correspondence and requests for materials should be addressed to B.A.V. (email: B.A.Verberne@uu.nl)

The largest upper-crustal earthquakes nucleate near the lower limit of the seismogenic zone, at depths ranging from 10 to 25 km depending on tectonic setting and local geotherm[1–5]. Beyond the seismogenic limit, slip is achieved via normal stress-insensitive, thermally activated, ductile flow, whereas at shallower depths this occurs via frictional processes in narrow, localized slip zones[1,2,6–10]. The transition from pure frictional slip to pure ductile flow with increasing depth is transitional, characterized by mixed-mode fault slip behaviour or "frictional-viscous" flow[11–13]. Understanding the transition from intrinsically stable ductile rock flow in shear zones to unstable frictional fault slip is widely recognized as a crucial development for improving our understanding of the processes controlling the depth to, and earthquake nucleation at, the lower boundary of the seismogenic zone[14–18].

Earthquakes are believed to nucleate on faults that exhibit self-enhancing, "velocity (v-) weakening" frictional behaviour, showing a decrease in fault strength with increasing displacement rate, as opposed to self-stabilizing v-strengthening behaviour[19,20]. Laboratory fault slip experiments have demonstrated transitions with increasing temperature from v-strengthening to v-weakening, and back to v-strengthening, broadly consistent with the upper-crustal seismogenic zone[21–24]. At low temperatures, shear strain is achieved via frictional granular flow in narrow, ultra-fine-grained shear bands[25,26]. Reference [26] showed that the upper transition, from v-strengthening to -weakening at shallow depths in the seismogenic zone can be explained by accelerated diffusive mass transfer rates in localized nanogranular slip zones, at least in simulated calcite gouges. However, at higher temperatures, thermally activated dislocation or diffusion creep processes in the bulk gouge play an increasingly important role, leading to a transition from v-weakening to -strengthening[23,24]. In layers of simulated calcite gouge sheared using sequentially stepped sliding velocities within the range from 0.03 to 100 μm s⁻¹, this transition occurred around 500–600 °C[23]. A slip stability transition at these high temperatures is suggestive of a change from brittle, frictional slip to ductile flow, similar to that observed with decreasing slip rate in sheared halite faults[7,12,27–30]. However, despite its importance in determining the base of the seismogenic zone, and the prevalence of large earthquakes in this depth range, the

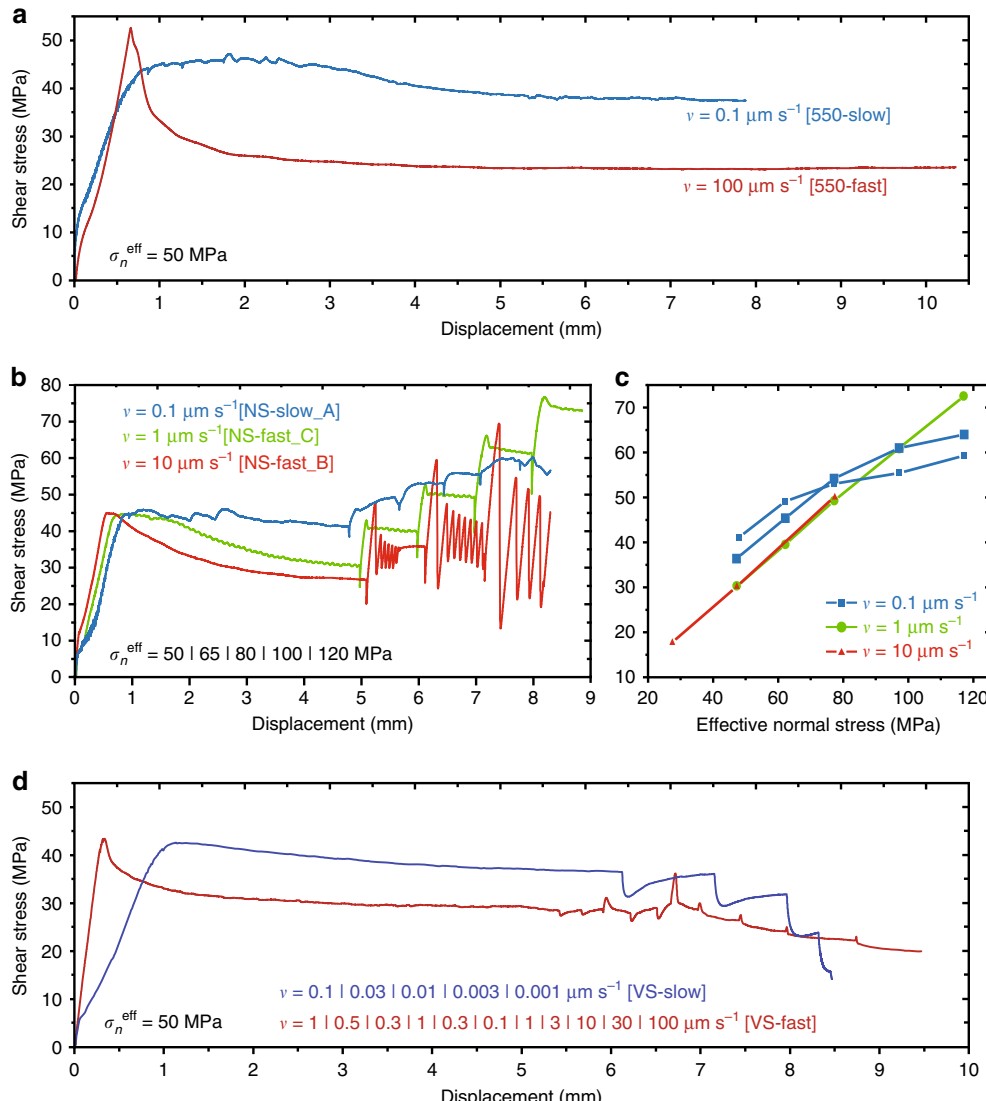

**Fig. 1** Hydrothermal shear tests on layers of simulated calcite fault gouge at 550 °C. **a** Experiments performed at a constant effective normal stress ($\sigma_n^{eff}$) of 50 MPa, using a sliding velocity (v) of 0.1 and 100 μm s⁻¹. **b** Effective normal stress-stepping experiments. **c** Steady-state shear stress plotted against effective normal stress, taken from experiments performed using v = 0.1 μm s⁻¹ (NS-slow_A and NS-slow_B) and ≥1 μm s⁻¹ (NS-fast_A and NS-fast_C). **d** Velocity-stepping experiments. Supplementary Tables 1–3 list all experiments and strength data

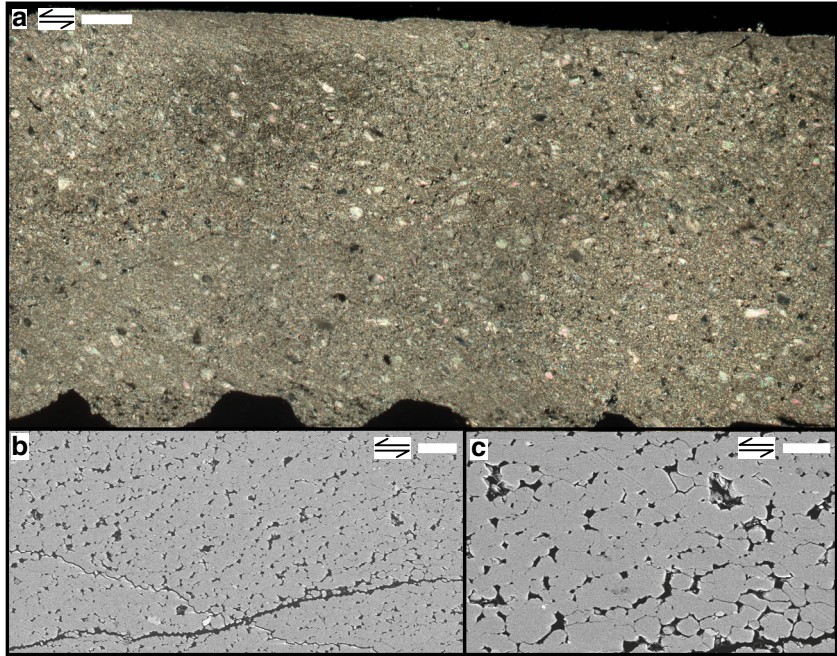

**Fig. 2** Microstructure of simulated calcite fault gouge sheared at 550 °C at 0.1 μm s$^{-1}$. Double arrows indicate sense of shear. **a** Cross-polarized light micrograph. Scale bar, 100 μm. **b**, **c** Backscatter electron micrographs, showing microstructures representative for the (relatively) homogeneously sheared gouge layer. Scale bars, 10 μm (**b**) and 5 μm (**c**)

microphysical processes controlling this transition remain enigmatic.

Here we report shear experiments on layers of simulated calcite fault gouge, at 550 °C, close to the transition temperature from $v$-weakening to $v$-strengthening[23]. Our results corroborate the importance of shear localization and porosity development close to the brittle-ductile transition[18,31–34], in a monomineralic, widespread crustal rock type. Using a microphysical model for slip of gouge-filled faults[35–37], we establish a mechanistic basis for the processes controlling earthquake nucleation close to the lower boundary of the seismogenic zone.

## Results

**Experiments**. Shear experiments were conducted using a hydrothermal ring shear apparatus, at ~550 °C, on ~1 mm thick layers of finely crushed (15–20 μm) Iceland Spar calcite crystals (Supplementary Table 1). In tests using a shearing velocity of 0.1 μm s$^{-1}$ and a constant effective normal stress ($\sigma_n^{eff}$) of 50 MPa (Fig. 1a), the shear strength vs. displacement curve shows rapid, near-linear loading in the first ~0.5 mm of displacement, followed by apparent yielding, hardening to a peak strength value, and gradual displacement weakening to a steady-state shear strength ($\tau_{ss}$) of 39 MPa. By contrast, at 100 μm s$^{-1}$, a sharply defined peak strength was reached after 0.7 mm of slip, followed by a rapid decay of shear strength until a steady-state value of ~23 MPa (Fig. 1a).

The effect of changing $\sigma_n^{eff}$-value on $\tau_{ss}$ was also investigated, increasing the effective normal stress within the range from 30 to 140 MPa, while employing constant sliding velocities of 0.1, 1 and 10 μm s$^{-1}$ (Fig. 1b, Supplementary Fig. 1). For each test, upon a step in effective normal stress, the shear stress initially increased rapidly. In tests conducted at 0.1 μm s$^{-1}$, this rapid increase of shear stress was followed by a gradual approach to stable, steady-state sliding, while for $v \geq 1$ μm s$^{-1}$, a sharply defined peak shear stress developed, followed by steady-state sliding or else stick-slip. A plot of values of steady-state shear strength (Supplementary Table 2) against $\sigma_n^{eff}$ shows a progressively decreasing slope for

experiments performed at 0.1 μm s$^{-1}$, but a positive, near-perfectly linear correlation for runs using $v \geq 1$ μm s$^{-1}$ (Fig. 1c).

To investigate the rate sensitivity of steady-state shear stress in our relatively slow ($v \leq 0.1$ μm s$^{-1}$) vs. fast ($v \geq 1$ μm s$^{-1}$) experiments, we performed $v$-stepping experiments, employing $\sigma_n^{eff} = 50$ MPa and initial sliding velocities of 0.1 and 1 μm s$^{-1}$ (Fig. 1d). For each step in sliding velocity, the shapes of the shear stress vs. displacement curves are consistent with those observed in constant-$v$ experiments (Fig. 1a), ie, showing a broad peak strength for $v \leq 0.1$ μm s$^{-1}$, but a sharply defined peak strength for $v \geq 1$ μm s$^{-1}$. For experiments using an initial sliding velocity of 0.1 μm s$^{-1}$, subsequent downward $v$-steps resulted in a decrease of the steady-state shear strength (Fig. 1d, Supplementary Table 3), pointing to $v$-strengthening behaviour. By contrast, using an initial sliding velocity of 1 μm s$^{-1}$, the steady-state shear strength value increased with decreasing displacement rate, and vice versa (Fig. 1d, Supplementary Table 3), implying $v$-weakening behaviour.

**Microstructures**. Gouge layers sheared at 0.1 μm s$^{-1}$ were usually recovered intact (Supplementary Fig. 2)[38]. Investigation of sectioned samples using light and electron microscopy revealed a relatively homogeneous microstructure consisting of 10–30 μm-sized rounded to elongated clasts dispersed in a fine dense matrix of ~1–5 μm-sized polygonal grains (Fig. 2). The grain size distribution is near-constant over the width of the sample layer, with a median ($\tilde{d}$) and mean ($\bar{d}$) grain size, respectively, measuring ~2.8 and 3.1 μm (Supplementary Fig. 3). Highly porous zones are frequently aligned at a low angle to the shear plane, and represent unloading or depressurization cracks formed upon cooling after the experiment (Fig. 2b, c, Supplementary Fig. 4). The lack of obvious shear bands suggests that shear strain was achieved through deformation of the entire ~1 mm thick sample layer, implying that the shear strain rate in the experiment measured $\dot{\gamma} = v/L \approx 1 \cdot 10^{-4}$ s$^{-1}$.

Samples sheared at $v \geq 1$ μm s$^{-1}$ split along inclined or boundary-parallel fractures whose surfaces frequently exhibit

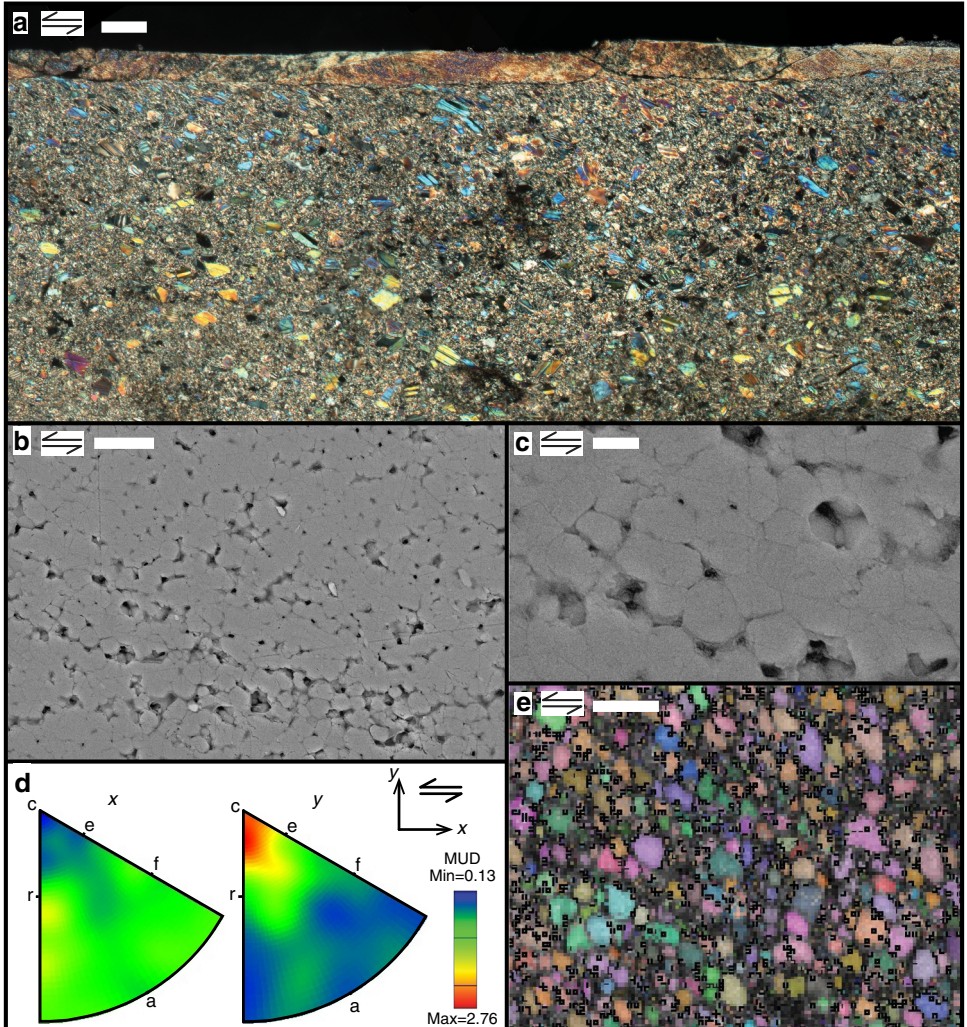

**Fig. 3** Microstructure of simulated calcite fault gouge sheared at 550 °C at 100 μm s$^{-1}$. Double arrows indicate sense of shear. **a** Cross-polarized light micrograph. Scale bar, 50 μm. **b**, **c** Backscatter electron micrographs taken within the shear band. Scale bars, 2 μm (**b**) and 500 nm (**c**). **d**, **e** Electron backscatter diffraction (EBSD) of shear band grains. Step size, 0.1 μm. **d** Upper hemisphere, equal area inverse pole figures plotting all EBSD data (7179 data points) in the Euler angle map of shear band grains shown in **e**. In **d**, the calcite crystallographic orientations are plotted with respect to an orientation parallel (x) and normal (y) to the shear direction. Scale bar, 2 μm. MUD = mean uniform density

slip-parallel striations and specular light reflection, implying splitting along narrow shear bands (Supplementary Fig. 2). The microstructure developed at 100 μm s$^{-1}$, at $\sigma_n^{eff} = 50$ MPa, shows a single ultra-fine-grained, boundary-parallel shear band, some 30–40 μm in width (Fig. 3a, Supplementary Fig. 4). This truncates the angular clasts comprising the bulk gouge, which otherwise resembles the starting material. The internal shear band grains are polygonal, ~0.3–1 μm in size and frequently arranged in linear, cavitated arrays (Fig. 3b, c). The shear band marks an abrupt reduction in grain size, which is typically interpreted to indicate that most of the imposed shear displacement was accommodated within it[39–41], suggesting an internal shear rate of $\dot{\gamma} \approx 2–3$ s$^{-1}$. The shear band further shows strong, uniform birefringence and optical extinction (Fig. 3a), suggestive of a crystallographic-preferred orientation (CPO). Electron backscatter diffraction (EBSD) data on the shear band grains reveal that the calcite {104} or r-plane and <$\bar{2}$01> direction are aligned subparallel to, respectively, the shear plane and shear direction (Fig. 3d, e)[38]. This preferred orientation is also present in the truncated, 10–30 μm-sized grains adjacent to the shear band (Supplementary Fig. 5).

## Discussion

The progressively decreasing slope seen in plots of steady-state shear stress against effective normal stress for experiments at 0.1 μm s$^{-1}$ points to more $\sigma_n^{eff}$-insensitive sliding behaviour, characteristic for frictional-viscous flow[7,12,27,28] (Fig. 1c). Microstructures of samples recovered from tests conducted at these slow rates show distributed shear, with porphyroclasts dispersed in a matrix of polygonal grains (Fig. 2, Supplementary Figs. 3 and 4), consistent with (dynamic) recrystallization and flow largely controlled by non-dilatant creep[42]. By contrast, effective normal stress-stepping experiments using $v \geq 1$ μm s$^{-1}$ showed a proportional increase of steady-state shear strength with increasing $\sigma_n^{eff}$-value, indicative of pure frictional slip, and a localized microstructure (Fig. 3, Supplementary Fig. 4). The granular texture seen in the shear band combined with the high shear strain rates operating here (up to 3 s$^{-1}$) imply that granular flow must have played a role in accommodating shear strain[23–25,29,35,40,41]. Thus, all mechanical and microstructural evidence points to a transition with increasing strain rate from a flow- to a friction-dominated shear regime.

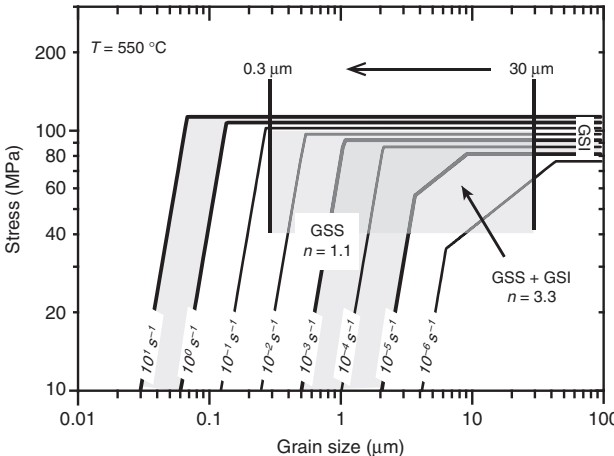

**Fig. 4** Deformation map for calcite at 550 °C. Constructed using the creep equations fitted by ref. [46] (grain size sensitive (GSS), $n = 1.1$) and ref. [43] (grain size insensitive (GSI) + GSS, $n = 3.3$) on data from axi-symmetric compression tests on synthetic calcite aggregates, and using an exponential-type, GSI creep equation fitted to data from triaxial compression test on Carrara marble by ref. [44]. Strain rate contours are shown for $\dot{\varepsilon} = 10^{-6}$ to $10^{1}\,\mathrm{s}^{-1}$. The shaded area plots the conditions relevant to our experiments, based on estimates from grain size and equivalent compressive stress

To investigate whether grain size insensitive (GSI, dislocation creep) and/ or grain size sensitive (GSS, diffusion creep) flow processes played a role in our experiments, we constructed a compressive flow stress vs. grain-size deformation map for calcite at 550 °C (Fig. 4). For our tests at 0.1 μm s$^{-1}$, the observed grain size of 1–30 μm, the equivalent flow stress ($\sigma = \tau\sqrt{3} \approx$ 65–80 MPa) and corresponding strain rate ($\dot{\varepsilon} = \dot{\gamma}/\sqrt{3} \approx 10^{-5}$ to $10^{-4}\,\mathrm{s}^{-1}$) are consistent with the ($\sigma$, $\dot{\varepsilon}$, $d$)-conditions needed for both GSS and GSI processes to operate. Using data from experiment VS-slow, conducted at $v \leq 0.1$ μm s$^{-1}$ (Fig. 1d), we calculated the stress exponent or "$n$-value" characteristic for constitutive equations describing flow in the diffusion and dislocation creep regimes[43–46]. We found progressively decreasing values with decreasing sliding velocities, from $n = 87$ at $v > 0.03$ μm s$^{-1}$ to $n = 2$ to $4$ at $v = 0.001$–$0.003$ μm s$^{-1}$. Very high $n$-values (ie, $\gg 10$) are frequently observed in gouge slip experiments conducted in the frictional regime[47,48], whereas $n$-values of 4–8 are characteristic for dislocation creep in calcite materials[45], and $n \approx 1$–$4$ for diffusion creep[45,46]. This suggests a gradual transition from frictional-viscous to fully viscous or ductile flow with decreasing strain rate. In view of the polygonal grain shape and mean grain size of ~3 μm (Fig. 2b, c, Supplementary Fig. 3), and the low $n$-values observed in the slow $v$-stepping experiment, our interpretation is that diffusion-accommodated grain boundary sliding (GBS) creep dominated deformation at low slip rates in our experiments.

For tests conducted using $v \geq 1$ μm s$^{-1}$, assuming a shear band developed in all runs similar to that at 100 μm s$^{-1}$ (Fig. 3), comparison of the grain size and equivalent stress and strain rate conditions suggests GSI and GSS processes alone cannot have accommodated the imposed shear strain. Instead, as mentioned above, the granular shear band texture combined with the high shear strain rates are strongly suggestive of (frictional) granular flow. On the other hand, the fine polygonal grain structure and CPO observed in the shear band strongly suggest that intergranular diffusion and/or dislocation creep played at least some role during slip. Significantly, the smallest grain size in the shear band (~0.3 μm) is close to the minimum grain size of ~0.4 μm expected from static recrystallization upon post-shear cooling[49,50]

(Supplementary Fig. 6), suggesting that diffusion creep has played an important role, with the grain size possibly being even finer during shear. The deformation map indicates that diffusion creep would dominate in the shear band for grain sizes of 30–90 nm (Fig. 4). For grains this small, the accompanying CPO may be explained by orientation-dependent sintering, as recently proposed for CPO development in sheared nanogranular aggregates[26,51]. This information, together with the linear, cavitated, polygonal grain arrays seen in the shear band, point to incompletely diffusion creep-accommodated GBS controlling shear band slip. However, dislocation creep-accommodated GBS, which is also capable of producing a CPO[52], cannot be excluded.

A transition with increasing shear strain rate from stable, creep-controlled flow to unstable, frictional sliding by dilatant granular flow is predicted by the microphysical model for shear of gouge-filled faults proposed by Spiers and co-workers[35–37] (Fig. 5). In the original model, shear strength is controlled by a competition between pressure solution creep and dilatational granular flow[29,35]. However, any rate-dependent, Arrhenius-type deformation mechanism will, when in competition with rate-insensitive GBS and dilatation, lead to potentially unstable, $v$-weakening fault slip[36,37]. This model can also explain our results for monomineralic calcite gouge, which transitions from non-dilatant, completely creep-accommodated GBS to dilatant, incompletely creep-accommodated GBS, causing $v$-weakening frictional slip. Intergranular dilatation, or cavitation, occurs because creep is not fast enough to accommodate GBS, leading to an increase in porosity, and to a balance between dilatation and compaction. This in turn leads to a strength reduction, producing $v$-weakening slip[35]. At the same time, porosity development leads to shear band localization[53,54], hence a further increase of the internal shear strain rate, and further velocity weakening. The shear instability thus established provides a mechanism of producing runaway slip, which can be enhanced further via the operation of dynamic weakening processes[55] such as frictional heating and thermal pressurization of pore fluid[56,57]. With trade-offs between temperature, slip velocity and effective normal stress[22,58], the stability transition seen in our experiments may occur under conditions consistent with plate motion at the base of the seismogenic zone.

Our findings are analogous to a ductile-to-brittle or flow-to-friction transition with increasing shear strain rate observed at room temperature in simulated halite and serpentinite faults[7,18,27–30,59], but provide the first data on, and clear mechanistic basis for, this transition, in a widespread, mono-mineralic crustal fault rock. While the transition from pure ductile to frictional-viscous flow is gradual, the switch to fully frictional slip, characterized by a linear dependence of shear stress on effective normal stress and velocity weakening behaviour, is abrupt. The key process controlling this switch from stable shear zone flow to frictional fault slip is grain-scale cavitation. The proposed mechanism is essentially a shear zone failure mechanism[31,32], where "failure" is defined as the onset of cavitation in the shear zone core[33,34], which triggers a shear instability leading to runaway fault slip. It occurs when creep processes cannot keep up with the shear strain rate, or when the shear stress becomes so high that creep is not fast enough to avoid cavitation. Since the deformation mechanisms observed to control the flow-to-friction transition in calcite operate in all rock-forming minerals across the brittle-ductile transition, we expect that our findings are generally applicable to faults in the upper crust. This means that under conditions near the base of the seismogenic zone, normally associated with ductile flow[60–65], transient changes in shear strain rate can trigger dilatant $v$-weakening frictional slip, with the capability of nucleating earthquakes.

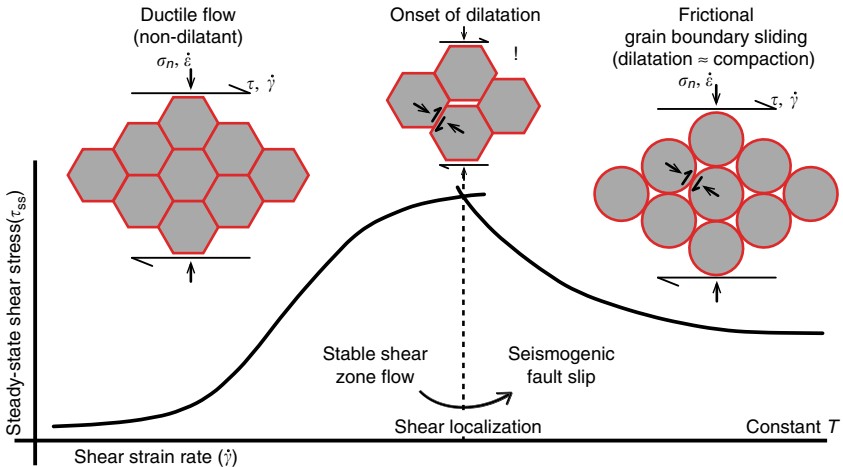

**Fig. 5** Representation of the transition from stable shear zone flow to seismogenic fault slip. Non-dilatant, ductile flow by creep-accommodated grain boundary sliding (GBS) transitions upon the onset of cavitation to a regime of incompletely creep-accommodated GBS and v-weakening frictional sliding. Cavitation occurs because creep processes become too slow to accommodate shear. For the original model formulation, see refs. [35-37]

## Methods

**Experiments and procedures**. Experiments were performed using the Utrecht hydrothermal ring shear apparatus[22,23,38]. In conducting each experiment, we distributed ~0.6 g of sample powder in the annular space between two grooved René-41 steel pistons, which we confined using an inner and an outer René-41 ring, with a diameter of, respectively, 22 and 28 mm. The confining rings are lubricated using Molykote D-321R anti-friction coating to reduce wall friction.

Each experiment was performed wet, using a furnace set point temperature of 550 °C and a pore fluid (demineralized water) pressure ($P_f$) of 100 MPa (Supplementary Table 1). Experiments that employed a fixed displacement rate ($v$) used a fixed effective normal stress ($\sigma_n^{\text{eff}}$) of 50 MPa, while effective normal stress-stepping tests were conducted employing $v = 0.1$, 1 and 10 µm s$^{-1}$ (Supplementary Table 1). A step in the $\sigma_n^{\text{eff}}$-value was achieved by first stopping the rotation drive, then increasing the axial load to the desired value, and, finally, re-starting rotation hence sample shear. This stepping procedure took around 1 min to complete. Velocity-stepping experiments were conducted using a constant effective normal stress of 50 MPa. To avoid the development of shear bands, the run which used an initial sliding velocity of 0.1 µm s$^{-1}$ was subject to downward velocity steps only. Upon terminating an experiment, we first stopped rotation, then switched off the furnace. The sample temperature follows an exponential decay to room temperature, falling from ~550 °C to ~280 °C in 115 s$^{38}$. After cooling to <100 °C, the vessel was depressurized, and the normal load was removed. In total, it took about 30 min between termination of the experiment and removal of the sample from the pressure vessel. In the velocity-stepping experiments, during cooling we maintained the pore fluid pressure above ~22 MPa, aiming to prevent a phase change of the pore fluid water from supercritical to liquid.

In the hydrothermal ring shear apparatus, the upper piston is pressure-compensated so that the effective normal stress equals the applied normal stress acting on the sample layer minus a contribution from the water-cooled internal O-ring seals, which measures ~1.9 MPa[38]. Shear stress was assumed uniform over the width of the sample layer (3 mm), and was corrected for seal friction using displacement- and pore-pressure-dependent calibrations[38]. Error propagation analysis showed that the error to the measured shear stress is ≤0.1%.

**Sample recovery, microscopes and microstructural analyses**. Sample fragments recovered after an experiment were impregnated using Araldite 2020 epoxy resin, and then left to cure for several days. Hardened samples were sectioned normal to the shear plane and tangential to a centrally inscribed circle[23]. All sections were polished to ~30 µm thickness, and investigated in transmitted light using a Leica polarizing light microscope[38]. Light microscope analysis of an unsheared gouge sample, pre-compressed at 50 MPa, showed the presence of ~10–30 µm-sized angular grains embedded in a matrix of ~1–5 µm-sized, angular to sub-rounded grains. A visually distinctive shape fabric or crystallographic texture was absent. Polished sections investigated using a scanning electron microscope (SEM) were sputter-coated with a ~5 nm thick layer of Pt/Pd to enable conduction. SEM imaging was done using a FEI Nova Nanolab 600 SEM or Helios Nanolab G3 SEM, operated in backscatter electron (BSE) mode, using an acceleration voltage of 5–7 kV and beam current of 0.2–0.5 nA. A selection of BSE micrographs was analysed for grain size distribution, manually delineating each visually distinctive grain with a polygon tracing tool (Supplementary Fig. 3). From the area of each traced grain we calculated the equivalent circular diameter as a proxy for grain size, which we then used for computing frequency histograms, ie, grain size distributions.

To investigate the crystallographic orientation distribution of the grains present in and close to the shear band developed at 100 µm s$^{-1}$ (Fig. 3d, e), we conducted

electron backscatter diffraction (EBSD) analysis. To this end, we first polished our sections further following the Syton method, ie, by re-polishing the sections using a silica colloid. After rinsing the Syton polished section with demineralized water, we applied a thin carbon coating. All EBSD measurements were carried out using an Oxford Instruments EBSD detector mounted on a FEI XL30S FEG SEM, mapping the crystallographic orientation of grains over areas measuring ~15 by 15 µm to 150 by 200 µm in size (Fig. 3e and Supplementary Fig. 5). All maps were recorded using an acceleration voltage of 25 kV, beam current ~2 nA, an aperture of 50 µm, at 20 mm working distance, using a step size of either 0.1 or 0.5 µm. The Kikuchi band pattern corresponding to each measurement or pixel is automatically indexed using Oxford Instruments AZtec software. Pixels at grain boundaries are frequently not indexed due to overlapping Kikuchi band patterns. Applying a large step size on a part of the section that is characterized by a relatively small grain size implies a high ratio of boundary-to-grain EBSD patterns, and a relatively low number of successfully indexed Kikuchi band patterns. All pole figures were constructed using a half width of 15° and cluster size of 5°.

**Construction of deformation map for calcite**. To construct the deformation map, we used empirical flow equations by ref. [46] for the GSS field, by ref. [43] for the GSS + GSI field, and using the experimental data of ref. [44] fitted to an exponential-type flow equation for the GSI field (see also ref. [23]). These flow laws were determined for axi-symmetric compression of dense calcite polycrystals at elevated pressure. We converted shear stress ($\tau$) and shear strain rate ($\dot\gamma$) into equivalent flow stress ($\sigma$) and strain rate ($\dot\epsilon$) for compression tests using the relations $\sigma = \tau\sqrt{3}$ and $\dot\epsilon = \dot\gamma/\sqrt{3}$. Taking $\tau \approx 23$ to 53 MPa as a broad range that represents the peak and steady-state shear stresses measured in our experiments using an effective normal stress of 50 MPa (Supplementary Tables 2 and 3), this results in an equivalent flow stress of $\sigma \approx 40$–90 MPa representing our samples in the deformation map. We estimated the shear strain rate in our experiments by dividing the imposed sliding velocity over the slip zone width, which we estimated from sectioned samples (Figs. 2 and 3). For experiment 550-slow, we assumed that the imposed displacement was either distributed over the entire ~1 mm thick layer, or that it was localized in a 50–100 µm-wide boundary shear. This yielded an end-member range for the internal shear strain rate of $\dot\gamma \approx 1 \cdot 10^{-4}$ s$^{-1}$ to $2 \cdot 10^{-3}$ s$^{-1}$, so that $\dot\epsilon = 6 \cdot 10^{-5}$ to $1 \cdot 10^{-3}$ s$^{-1} \approx 10^{-5}$ to $10^{-3}$ s$^{-1}$. For experiment 550-fast, the imposed displacement was accommodated within a 30–40 µm-wide zone, so that $\dot\gamma \approx 2.5$–3.3 s$^{-1}$, and $\dot\epsilon = 1.4$–1.9 s$^{-1} \approx 10^{0}$–$10^{1}$ s$^{-1}$. To represent these values of $\dot\epsilon$ in the deformation map, we plotted lines of constant strain rate for the range $\dot\epsilon \approx 10^{-6}$–$10^{1}$ s$^{-1}$. We included $\dot\epsilon = 10^{-6}$ s$^{-1}$ because this more closely approximates the strain rates used in the experiments on which the flow equations are based[43,45,46].

**Grain growth due to static recrystallization upon cooling**. Grain growth after termination of our shear experiments may have played a role. Static grain growth of grains of initial size $d_0$ to a new size $d$ is frequently described using a generalized Arrhenius expression given by[49,50]

$$d^{1/n} - d_0^{1/n} = k_0 t \exp(-Q_g/RT), \tag{1}$$

where $k_0$ is a constant, $t$ is the duration of the growth period, $Q_g$ is the (apparent) activation energy for the process controlling the grain growth rate, $R$ is the gas constant and $T$ is the temperature during grain growth. Post-test, static recrystallization of our samples may have occurred upon termination of a test, during cooling from the testing temperature to room temperature conditions. Measurement of part of this temperature decay upon termination of experiment 550-fast[38]

showed a decrease from 550–280 °C in 115 s. Taking $\dot{T} = -c \cdot (T - T_a)$, where $T$ (0) = 550 °C, $T$(115) = 280 °C and $T_a$ = 20 °C, the decay rate can be approximated by

$$T(t) \approx 20 + (550 - 20)\exp(-ct), \text{ with } c = \frac{1}{115}\ln\left(\frac{280 - 20}{550 - 20}\right). \quad (2)$$

Using equations 1 and 2, plus kinetics parameters for surface-diffusion-controlled, static growth of calcite grains[49], we simulated the grain size that may have developed in the first 110 s of cooling of our samples, for initial grain sizes ranging from $d_0$ = 0.01–0.6 μm in size (Supplementary Fig. 6). This shows that, regardless of the initial size $d_0$, the bulk of grain growth will have been achieved within the first ~20–25 s of cooling. Also, for grains with an initial size smaller than ~0.25 μm, the final size is more or less constant at $d \approx 0.4$ μm (cf. Fig. 3), whereas for grains with $d_0$ exceeding 0.25 μm, the total growth in size $(d–d_0)$ will be limited to <0.1 μm. Allowing for differences in the detailed diffusion mechanisms[49], the minimum grain size found in our samples thus roughly corresponds with the "minimum" grain size as predicted by diffusion-controlled grain growth.

**Data availability**. All data that support the findings of this study are available from ref. [38]. Microscope images in their original resolution are available on request from the corresponding author.

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

## Acknowledgements

We thank O. Stiekema, L. Bik, T. van der Gon-Netscher, G. Kastelein, P. van Kriekena and E. de Graaff for technical support. B.A.V. was supported by grant 2011-75 awarded under the Netherlands centre for Integrated Solid Earth Science (ISES) and A.R.N. and J.C. by ERC starting grant SEISMIC (no. 335915), and by the Dutch Foundation for Scientific Research (NWO) through a VIDI grant (no. 854.12.011).

## Author contributions

B.A.V. and A.R.N. conceived the idea and designed the experiments. B.A.V. carried out the experiments and microscope work (except EBSD), and analysed the data. J.C. conducted the *v*-stepping experiments, J.H.P.d.B. helped with the deformation map and G.M.P. carried out the EBSD analyses. J.H.P.d.B., G.M.P. and M.R.D. helped interpreting the EBSD data. B.A.V., A.R.N. and C.J.S. co-wrote the manuscript, while C.J.S. supervised the whole project. All authors discussed the results and checked the manuscript.

## Additional information

**Competing interests:** The authors declare no competing financial interests.

