## [Peer Review File · Nature Communications]

Reviewers' comments:

Reviewer #1 (Remarks to the Author):

The topic area and the findings and conclusions have the potential to be significant and timely. The key findings and conclusions of the manuscript are clearly summarized by the statements in the abstract, specifically "Here we show that the flow-to-friction transition in experimentally simulated calcite faults is characterized by a switch from non-dilatant, dislocation and diffusion creep to dilatant deformation, involving incompletely accommodated grain boundary sliding," and also the statement applying the results to earthquake faulting, specifically, "The observed shear instability, triggered by the onset of microscale cavitation, provides a key mechanism for bringing about the brittle-ductile transition and nucleating (major) earthquakes at depths normally associated with ductile flow." Unfortunately, after reading the manuscript I do not think the authors have made a convincing case to warrant such statements. I do not think the data and analyses support the conclusions presented in the paper.

The main observations based on laboratory experiments of shearing granular calcite at high pressure and temperature appropriate to earthquake faulting is the documentation of two distinct modes of shearing behavior, stable shearing at low velocity and unstable (stick-slip) shearing at higher velocity, and that these modes correlate with distributed deformation and very localized deformation, respectively. However, the experimental data summarized in Figure 1c clearly and unequivocally shows that for all the conditions tested the behavior is frictional. That is, for both modes of behavior, the shear strength increases with increase in effective normal stress, and the dependence is quite pronounced consistent with friction. Moreover, the stick slip behavior showing increase in magnitude of stick-slip stress drops with increasing effective normal stress is a classic characteristic of frictional instability.

Microstructures in both regimes are consistent with dilatant, frictional behavior. Microstructures documented for both the distributed-stable shear regime and the localized-unstable shear regime show granular textures with intergranular porosity, and in fact, the images show what appears to be a rather large absolute porosity in both regimes (Figure 2, and Figure 3 in the extended data). Such textures and frictional mechanical behavior is associated with dilatancy, and there is no proof for absence of dilatancy. Thus, the first statement quoted from the abstract above, that the distributed-stable shearing regime reflects non-dilatant, dislocation and diffusion creep, is not substantiated. The authors also state, in lines 108-110, that "These mechanical and microstructural observations point to a transition from oneff-insensitive, ductile flow at low shear rates ($\leq 0.1 \mu\text{m/s}$) to linearly oneff-dependent, frictional sliding at high shear rates ($\geq 1 \mu\text{m/s}$)." I think the evidence marshaled for inferring diffusion and dislocation processes are important is correct, but not about non-dilatant and non-frictional at low velocities. Also, given that both regimes display intergranular porosity, the second statement quoted from the abstract above, that shear instability is triggered by the onset of cavitation, is also unfounded because cavitation (evidenced by porosity) is present in both regimes.

Much past experimental work on friction has documented stability transitions with changes in sliding rate, temperature, and normal stress, as well as changes in the steady-state rate dependence (a-b) with normal stress, temperature and velocity. These works commonly show that for rate and state friction, these transitions can result from progressive but differing changes in the magnitude of the a parameter and the b parameter with velocity, temperature or normal stress. Also, such transitions often, though certainly not necessarily, correlate with changes in shear localization where rate weakening and propensity for instability is correlated with shear localization. Noteworthy in this respect is prior work on halite by several workers including some of these authors. Although the results reported in the paper are interesting, they are not particularly remarkable or new.

In terms of the experiments, overall they are well done. I do have some minor concern whether

pore fluid pressure is able to be maintained in the deforming calcite layers under all conditions of testing, and even if it is, what the effective stress law is for deformation involving changes in porosity, intergranular sliding and dislocation/diffusion processes. This study begs for some additional experimental work to test effective stress with different combinations of pore fluid pressure and confining pressure, as well as to actually measure rate dependence of friction directly via velocity stepping.

To some extent, for this paper to focus on frictional instability in terms of rate weakening and dilatancy (or not) without directly measuring either, is problematic. The statement in the abstract that onset of microscale cavitation can nucleate major earthquakes could be correct in some cases, but certainly is not supported by the observations in this paper. Accordingly I cannot recommend this paper for publication.

Reviewer #2 (Remarks to the Author):

The manuscript presents laboratory experiments that aim to clarify the nature of shear resistance at depths with relatively high temperatures (~ 550 C) that are typically thought to be stable. The experimental results are interesting and novel, and suggest complex behaviors at different slip rates and effective stresses, including transition from distributed shear and stable slip to shear localization and weakening. They would be of interest to a wide range of scientists working on earthquakes. Hence the work is in principle suitable for publication in Nature Communications. At the same time, there are a number of unclear points that require major changes to the text and perhaps additional experiments.

1. The claim in the title of the manuscript, "Microscale cavitation as a mechanism for nucleation of major earthquakes," does not seem to be supported by the work and discussion in its present form. Briefly, (i) the results presented do not suggest that the layers studied can accelerate from locked to seismic slip rates, the process typically called "nucleation" and (ii) even if such nucleation could spontaneously happen, I am not sure why this process would necessarily nucleate "major earthquakes;" I am not aware of any studies that suggest that any large upper-crustal earthquakes have nucleated in the locations with the relatively high (550 C) temperatures considered in the study. Here are some more detailed thoughts on these two points.

(i) Before earthquake nucleation at the bottom of the seismogenic zone, the region in question is likely locked, or perhaps creeping with slip rates comparable to the loading plate rates. The loading plate rates are in tens of millimeters per year and correspond to about $10^{(-9)}$ m/s. According to the experiments, slip at slip rates of 0.1 micrometer/s = $10^{(-7)}$ m/s or lower is stable. Then how would a region at the bottom of the seismogenic zone, creeping with plate-like velocities of $10^{(-9)}$ m/s or lower (if it is partially locked) self-accelerate to nucleate dynamic rupture, if such slip velocities are stable? And if another mechanism is required to start the nucleation and bring it to the slip rates of 1-100 microns/s = $10^{(-6)}$ - $10^{(-4)}$ m/s that are shown to be unstable in this work, then why not attribute the nucleation to that other mechanism?

(ii) The bottom of the upper crustal seismogenic zones capable of spontaneously nucleating unstable frictional slip is typically associated with temperatures of 250-400 C. If the authors are aware of many (any?) major earthquakes nucleating in regions with the temperature of 550 C, they should cite these examples. Furthermore, if rupture nucleates at any depth, it is much more likely to become a microearthquake than a "major" earthquake, simply because there are a lot more small events nucleating at any depths. So even if the process of microscale cavitation discussed in the manuscript could lead by itself to earthquake nucleation, it is unclear why that would be a mechanism for nucleation of major earthquakes specifically, and not any earthquakes more generally. In fact, the transition from stable to unstable behavior described in the manuscript would be more relevant to observations of microseismic aftershocks after major events that occur deeper than interseismic microseismicity, presumably because major events increase creeping rates at depth.

2. There are several unclear points about the experimental results and their interpretation.

(i) Why is the temperature of 550 C used? As explained above, it seems to be too high to be the most relevant, so it is unclear why this and only this temperature is considered. It would help if the manuscript contained some discussion of the thermal structure of the relevant regions, why the temperature of 550 C was chosen, and why other temperatures are not considered. It would be even more illuminating to see some experiments at lower temperatures.

(ii) How are the "steady-state" shear stress values in Figure 1c are determined and what is their uncertainty/error? For some cases, like the effective stresses of 90 and 100 MPa in Figure 1b, stick-slip occurs, so there is no steady-state sliding and hence the "steady-state" shear stress is not obvious. For the case of 50 MPa effective stress in Figure 1b, the shear stress seems to take multiple values, between 25 and 30 MPa in the beginning of the experiment (at about 5 mm slip) and not more than 15 MPa by the end of the experiment (judging by the results for the closely related 60 and 40 MPa for 10-12 mm slips), clearly indicating that the process is quite complex and path-dependent, and there is no single steady-state shear stress value for the effective stress of 50 MPa and slip rate of 10 microns/s. Yet Figure 1c contains unambiguous steady-state shear stress values for all these case; for example, a value slightly higher than 30 MPa is used for the case of 50 MPa effective stress, which is not at all supported by the values from Figure 1b.

(iii) As slip accumulates in the experiment of Figure 1b, the fault gets progressively weaker, as evident from the discussion of the 50 MPa effective stress above. Why is that? Is this the effect of local shear heating and perhaps pore fluid pressurization?

(iv) The results for the slip rate of 0.1 micrometer/s in Figure 1c are described as "ductile flow" "insensitive" to the effective stress (e.g., lines 108-109). Yet the shear resistance goes up significantly with the effective stress in Figure 1c, just not as significantly as for higher slip rates. What is this due to? Does this mean that the behavior for slip rates of 0.1 micrometer/s is a mixture of ductile flow and frictional sliding, not just ductile flow as stated in the manuscript?

(v) Is dilatancy measured in the experiments? If yes, it would be good to report it. If not, why not?

3. Minor comments.

(i) Line 34 and similar occurrences: "velocity weakening" is an adjective quantifying frictional behavior, and hence could use a dash: "velocity-weakening." In contrast, no dash is fine in line 41, where weakening is a noun.

(ii) A number of people are thanked in the acknowledgements, but it is not specified for what.

Reply to Reviewers' Comments

Original submission date: 13 March 2017

Manuscript #: NCOMMS-17-06190-T

We thank the editor and reviewers for their assessment of our manuscript (ms) entitled '*Microscale cavitation as a mechanism for the nucleation of major earthquakes*'. We have revised the ms on the basis of the constructive comments made by the reviewers, including additional experimental and microstructural data, and rewriting/ reorganizing the main text, figures, as well as the supplement. The revised ms has lengthened, however it has improved significantly in clarity and content, while remaining well within the guidelines for a Nature Communications *Article*. Below we specify how we have improved the ms, addressing each of the points made by the reviewers.

Reviewers' comments

Reviewer #1 (Remarks to the Author):

The topic area and the findings and conclusions have the potential to be significant and timely. The key findings and conclusions of the manuscript are clearly summarized by the statements in the abstract, specifically “Here we show that the flow-to-friction transition in experimentally simulated calcite faults is characterized by a switch from non-dilatant, dislocation and diffusion creep to dilatant deformation, involving incompletely accommodated grain boundary sliding,” and also the statement applying the results to earthquake faulting, specifically, “The observed shear instability, triggered by the onset of microscale cavitation, provides a key mechanism for bringing about the brittle-ductile transition and nucleating (major) earthquakes at depths normally associated with ductile flow.” Unfortunately, after reading the manuscript I do not think the authors have made a convincing case to warrant such statements. I do not think the data and analyses support the conclusions presented in the paper.

We thank the reviewer for recognizing the potential impact of our findings and conclusions. In the revised ms we have addressed the reviewers concerns by providing additional experimental data, as well as by rewriting parts of the main text (see our replies below).

The main observations based on laboratory experiments of shearing granular calcite at high pressure and temperature appropriate to earthquake faulting is the documentation of two distinct modes of shearing behavior, stable shearing at low velocity and unstable (stick-slip) shearing at higher velocity, and that these modes correlate with distributed deformation and very localized deformation, respectively. However, the experimental data summarized in Figure 1c clearly and unequivocally shows that for all the conditions tested the behavior is frictional. That is, for both modes of behavior, the shear strength increases with increase in effective normal stress, and the dependence is quite pronounced consistent with friction.

Moreover, the stick slip behavior showing increase in magnitude of stick-slip stress drops with increasing effective normal stress is a classic characteristic of frictional instability.

We agree with the reviewer that for all experiments summarized in (revised) Fig. 1c, the steady-state shear stress (τ_{ss}) increases with increasing effective normal stress (σ_n^{eff}), suggestive of frictional behaviour. However, the curve to the data from runs at $v = 0.1 \mu\text{m/s}$ clearly flattens-off towards higher effective normal stresses, while for $v \geq 1 \mu\text{m/s}$ we find a positive, near-perfectly linear correlation. This difference in slope of linear versus flattening-off in experiments at relatively high versus low displacement rates strongly suggests a transition with decreasing strain rate from a friction-dominated regime to a flow-dominated regime (cf. e.g. *Shimamoto, 1986; Kawamoto & Shimamoto, 1997, 1998*).

To address the reviewers comment, in the revised ms we have clarified the presentation and interpretation of the effective normal stress stepping data, including references to classic works showing a similar friction-to-flow transition in halite (see e.g. lines 143-155 of the revised ms). Below we provide a detailed list of changes made to the ms, addressing this comment as well as other comments made by the reviewer.

As for the comment on stick-slip behaviour as a classical characteristic of frictional instability, we fully agree. Since we argue that (potentially unstable) velocity weakening frictional behaviour occurs in runs using $v \geq 1 \mu\text{m/s}$, but stable velocity strengthening flow in runs using $v \leq 0.1 \mu\text{m/s}$, our interpretations are in fact fully consistent with the reviewers point. Indeed, frictional instability or stick-slip only occurred in tests using $v \geq 1 \mu\text{m/s}$ (revised Fig. 1 and Supplementary Table 2).

Microstructures in both regimes are consistent with dilatant, frictional behavior. Microstructures documented for both the distributed-stable shear regime and the localized-unstable shear regime show granular textures with intergranular porosity, and in fact, the images show what appears to be a rather large absolute porosity in both regimes (Figure 2, and Figure 3 in the extended data). Such textures and frictional mechanical behavior is associated with dilatancy, and there is no proof for absence of dilatancy.

OK, point taken. We agree with the reviewer that all recovered microstructures show granular textures with intergranular porosity. However, crucially, in experiments in the frictional regime slip is localized in a narrow band, while in the flow regime, slip is distributed over the entire sample layer. The high shear strain rates applying to a porous, granular shear band (up to 3 s^{-1} in our experiments) strongly suggests dilatant granular flow played a role, consistent with frictional sliding (cf. e.g. *Verberne et al., 2013, 2015; Smith et al., 2015; Rempe et al., 2017*). However, in the flow regime, the much lower shear strain rates ($\sim 10^{-4} \text{ s}^{-1}$) combined with a microstructure composed of porphyroclasts dispersed in a matrix of polygonal grains, is consistent with non-dilatant, creep-controlled flow. This is supported by a comparison with the deformation map of calcite.

To address the reviewers comment, we have clarified our description and interpretation of the mechanical and microstructural data, using explicit references supporting our case. Below we provide a list of revisions specifically addressing this and other points made by the reviewer.

Thus, the first statement quoted from the abstract above, that the distributed-stable shearing regime reflects non-dilatant, dislocation and diffusion creep, is not substantiated. The authors also state, in lines 108-110, that “These mechanical and microstructural observations point to a transition from σ_{eff} -insensitive, ductile flow at low shear rates ($\leq 0.1 \mu\text{m/s}$) to linearly σ_{eff} -dependent, frictional sliding at high shear rates ($\geq 1 \mu\text{m/s}$).” I think the evidence marshaled for inferring diffusion and dislocation processes are important is correct, but not about non-dilatant and non-frictional at low velocities. Also, given that both regimes display intergranular porosity, the second statement quoted from the abstract above, that shear instability is triggered by the onset of cavitation, is also unfounded because cavitation (evidenced by porosity) is present in both regimes.

It is important to emphasize that the transition from pure friction to pure ductile flow in sheared gouge is characterized by mixed-mode or ‘frictional-viscous’ flow behaviour (e.g. *Handy et al., 1999; Bos & Spiers, 2002; Holdsworth et al., 2001; Imber et al., 2008*). Therefore, close to the transition, as is the case in our experiments, a frictional ‘component’ of shear is to be expected. While the transition from pure ductile to friction-viscous flow is gradual, the switch to fully frictional slip, characterized by a near-perfectly linear dependence of shear stress on effective normal stress and velocity weakening behaviour, is abrupt (see also Fig. 6 in the revised ms). This switch occurs because intergranular dilatation or grain-scale cavitation facilitates localization and granular flow, triggering a positive feedback mechanism. Any intergranular dilatation in the frictional-viscous flow regime occurs in insignificant amounts, such that time-dependent deformation processes are fast enough to (sufficiently) compensate and prevent localization or the triggering of an instability.

Multiple lines of evidence are in support of our interpretation of non-dilatant, creep-controlled flow at low rates vs dilatant, frictional slip at high rates in our experiments;

1. Microstructures. Samples sheared at low rates in our experiments show evidence for distributed shear, with porphyroclasts in a matrix of polygonal grains (revised Fig. 2). This is consistent with non-dilatant, creep-controlled flow involving (dynamic) recrystallization (*Fliervoet et al., 1997*). By contrast, at high slip rates we found localized slip in a granular shear band (revised Fig. 3), strongly suggesting that (dilatant) granular flow controlled shear strain accommodation (*Verberne et al., 2013, 2015; Smith et al., 2015; Rempe et al., 2017*)
2. Effective normal stress stepping data. The progressively decreasing slope seen in plots of steady-state shear stress against effective normal stress for experiments at $0.1 \mu\text{m/s}$ points to σ_n^{eff} -insensitive sliding behaviour, characteristic for frictional-viscous flow
3. Calcite deformation maps. Comparison of grain size determined from recovered microstructures and estimates of equivalent compressive stress and strain rate based on our experimental data, with deformation maps for calcite (revised fig 4), strongly suggest that dislocation/ diffusion creep played a role in the low velocity (‘flow regime’) experiments. However, in the high velocity experiments, the shear strain rates applicable to shear band slip are too high.
4. Velocity stepping data. Experiment VS-slow, using $v \leq 0.1 \mu\text{m/s}$, shows intrinsically stable velocity strengthening behaviour, as expected for creep-controlled flow, whereas experiment VS-fast, using $v \geq 1 \mu\text{m/s}$, shows velocity weakening implying frictional sliding. Furthermore, the data from experiment VS-slow demonstrates the gradual nature

of the transition from frictional-viscous to fully ductile flow. The stress exponent or “*n*-value”, calculated our *v*-stepping data (Supplementary Discussion of the revised ms), shows progressively decreasing values from $n = 87$ at $v > 0.03 \mu\text{m/s}$, to $n = 2$ to 4 at $v = 0.001$ to $0.003 \mu\text{m/s}$. High *n*-values are typically observed in gouge slip experiments conducted in the frictional regime (Mitchell et al., 2013; Chen et al., 2015), whereas *n*-values of 2 to 4 are characteristic for deformation by dislocation or diffusion creep-controlled flow (De Bresser et al., 2002; Herwegh et al., 2003).

Thanks to the reviewers comments we now realize that in the original ms we had not sufficiently described the lines of evidence described above. To improve this, and other points made by the reviewers, we have revised the ms in the following way;

- i. We have performed new experiments. These additional data on the velocity dependence of sliding in the flow vs. frictional regime strengthen our interpretations (point 4 above). See Fig. 1d, Suppl. Table 1 and 3, Suppl. Fig. 4, and lines 79-88, 162-173, 264-267, 274-276 of the revised ms.
- ii. We have rewritten the main text. We now more explicitly describe our interpretations, including on the gradual nature of the transition from pure ductile to frictional-viscous flow, and on the role of dilatancy in controlling the switch to frictional slip. We use references to relevant previous works to support our interpretations where necessary, and provide more background to explain our rationale and approach. See lines 11-23 (abstract), 25-58 (introduction), 143-155, 162-173, 185-189, 230-234 (discussion) of the revised ms.
- iii. We have reorganized presentation of mechanical data. The effective normal stress stepping data now only contain upward steps in effective normal stress. This declutters the data, while there is no essential information lost. We further improved clarity by referring to experiment names within the figure, as listed in Supplementary Table 1, and we more explicitly describe the data in the main text. Lastly, we now provide shear stress vs displacement plots of all experiments performed, either in the main text or in the Supplement. See Fig. 1b,c, Suppl. Fig. 1, and lines 61-88 of the revised ms.
- iv. We have reorganized figures. In addition to changes made to Fig. 1 (see previous point), we have split the microstructural data into one for the slow and one for the fast experiments. One of the advantages of this is that we can now incorporate EBSD data demonstrating the shear band CPO into the main text, which is quite a technological achievement on such small grains, as well as a rather unique observation. Further, in the revised ms we incorporate the calcite deformation map into the main text, because it convincingly shows the importance of creep-controlled flow in our slow experiments (point 3 above). See Figs. 2, 3, 4 of the revised ms.
- v. We added more microstructural evidence to the Supplement. We now use Supplementary Fig. 4 to show lower magnification BSE images of slow versus fast deformed samples. This provides more evidence for the differences between these experiments, but also helps to show depressurization cracks developed in the slow experiments (lines 104-107 of the revised ms).

Much past experimental work on friction has documented stability transitions with changes in sliding rate, temperature, and normal stress, as well as changes in the steady-state rate

dependence (a-b) with normal stress, temperature and velocity. These works commonly show that for rate and state friction, these transitions can result from progressive but differing changes in the magnitude of the a parameter and the b parameter with velocity, temperature or normal stress. Also, such transitions often, though certainly not necessarily, correlate with changes in shear localization where rate weakening and propensity for instability is correlated with shear localization. Noteworthy in this respect is prior work on halite by several workers including some of these authors. Although the results reported in the paper are interesting, they are not particularly remarkable or new.

Our findings show that localization is crucial in controlling slip stability in calcite at high temperatures. We consider the internal consistency of the present microstructural and mechanical data, as well as with the Spiers model and previous works on halite, quite remarkable. Note that previous studies have not explicitly invoked on the deformation mechanisms active during shear of a widespread, monomineralic crustal fault rock. Moreover, since the deformation mechanisms observed to play a role in our experiments are relevant to any crustal fault rock undergoing a ductile-to-brittle transition, our inferences are expected to be generally applicable (as noted in the concluding paragraph).

To address the reviewers comment, we expanded the introduction to better place our work in relevant context. We now explicitly mention relevant previous work on shear mechanisms at the lower boundary of the seismogenic zone, including on ductile failure (*Shigematsu et al., 2004; Rybacki et al., 2008*) and cavitation (*Fusseis et al., 2009; Menegon et al., 2015*), and recent work hinting on the key role of localization to earthquake nucleation at the lower boundary of the seismogenic zone (*Takahashi et al., 2017*). See lines 53-58 and 227-243 of the revised ms.

In terms of the experiments, overall they are well done. I do have some minor concern whether pore fluid pressure is able to be maintained in the deforming calcite layers under all conditions of testing, and even if it is, what the effective stress law is for deformation involving changes in porosity, intergranular sliding and dislocation/diffusion processes.

While the reviewer has a point concerning the effective normal stress state in for that matter in any gouge sliding experiment that possibly involve a low dynamic permeability (e.g. *Morrow et al., 2000; Verberne et al., 2013*), we expect that this is not a major problem in our experiments. Differences in pore pressure will dissipate, or else steady-state is not likely to be achieved. Since all experiments not showing stick-slip showed near-steady state sliding behaviour, we are confident that our shear stress data are trustworthy. Moreover, as mentioned above, multiple independent lines of evidence support our interpretations, which further strengthens our case.

This study begs for some additional experimental work to test effective stress with different combinations of pore fluid pressure and confining pressure, as well as to actually measure rate dependence of friction directly via velocity stepping.

We share the reviewers' curiosity for doing additional experiments, especially on the rate dependence of strength. We thank the reviewer for pointing this out. Indeed, as mentioned above, we have added v -stepping experiments to the revised ms, one using $v \leq 0.1$

$\mu\text{m/s}$ (VS-slow) and one using $v \geq 1 \mu\text{m/s}$ (VS-fast). See Fig. 1d and lines 79-98, 162-173, 264-267 of the revised ms.

On using different combination of pore fluid pressure and confining pressure, we agree that this is also interesting, however there are limits to what one could and should present in a single manuscript. Moreover, variations in confining pressure cannot be easily tested using the Utrecht Ring shear apparatus (see *Den Hartog et al., 2012; Verberne et al., 2015*).

To some extent, for this paper to focus on frictional instability in terms of rate weakening and dilatancy (or not) without directly measuring either, is problematic. The statement in the abstract that onset of microscale cavitation can nucleate major earthquakes could be correct in some cases, but certainly is not supported by the observations in this paper. Accordingly I cannot recommend this paper for publication.

We thank the reviewer for his/ her constructive comments, which we feel have helped to improve the significantly.

Reviewer #2 (Remarks to the Author):

The manuscript presents laboratory experiments that aim to clarify the nature of shear resistance at depths with relatively high temperatures (~550 C) that are typically thought to be stable. The experimental results are interesting and novel, and suggest complex behaviors at different slip rates and effective stresses, including transition from distributed shear and stable slip to shear localization and weakening. They would be of interest to a wide range of scientists working on earthquakes. Hence the work is in principle suitable for publication in Nature Communications. At the same time, there are a number of unclear points that require major changes to the text and perhaps additional experiments.

We thank the reviewer for recognizing the potential impact of our work, and its suitability for publication in Nature Communications.

1. The claim in the title of the manuscript, “Microscale cavitation as a mechanism for nucleation of major earthquakes, “ does not seem to be supported by the work and discussion in its present form. Briefly, (i) the results presented do not suggest that the layers studied can accelerate from locked to seismic slip rates, the process typically called “nucleation” and (ii) even if such nucleation could spontaneously happen, I am not sure why this process would necessarily nucleate “major earthquakes;” I am not aware of any studies that suggest that any large upper-crustal earthquakes have nucleated in the locations with the relatively high (550 C) temperatures considered in the study.

It is well-established that seismogenesis occurs on faults showing velocity weakening behaviour (e.g. *Scholz, 1998*). By studying the microphysical processes controlling the onset of velocity weakening under conditions pertaining to the lower boundary of seismogenic zone, we study key processes relevant to earthquakes nucleating there. Since the strongest earthquakes frequently nucleate close to the lower boundary of the seismogenic zone (lines 25-27 of the revised ms), we suggest our work is relevant to the nucleation of major earthquakes. We chose 550°C to simulate a friction-to-flow transition in sheared calcite layers, as previously observed to occur around 500 to 600°C in calcite gouge layers sheared under similar conditions of effective normal stress and fluid pressure (*Verberne et al., 2015*).

To address the reviewers comment, in the revised ms provide more background on the importance of the velocity dependence of shear strength in controlling fault stability (lines 35-40), and we better explain our rationale in choosing the experimental conditions (lines 44-52). Also, we provide new data on the velocity dependence of strength, consistent with all other results (see also replies to comments made by Reviewer #1).

Here are some more detailed thoughts on these two points.

(i) Before earthquake nucleation at the bottom of the seismogenic zone, the region in question is likely locked, or perhaps creeping with slip rates comparable to the loading plate rates. The loading plate rates are in tens of millimeters per year and correspond to about 10^{-9} m/s. According to the experiments, slip at slip rates of 0.1 micrometer/s = 10^{-7} m/s or lower is stable. Then how would a region at the bottom of the seismogenic zone, creeping with plate-like velocities of 10^{-9} m/s or lower (if it is partially locked) self-accelerate to nucleate dynamic rupture, if such slip velocities are stable? And if another mechanism is required to

start the nucleation and bring it to the slip rates of 1-100 microns/s = $10^{(-6)} - 10^{(-4)}$ m/s that are shown to be unstable in this work, then why not attribute the nucleation to that other mechanism?

The reviewer raises an interesting point here. We suggest that a region at the bottom of the seismogenic zone creeping at plate-like velocities may self-accelerate to the velocity-weakening regime, through localization, implying that the processes causing localization are crucial in triggering an instability (e.g. recrystallization, reaction-weakening – *Wintsch et al., 1995; Jin & Karato, 1998*). However, the key process for bringing about a shear instability is grain-scale cavitation, since this facilitates granular flow and velocity weakening, triggering the positive-feedback mechanism or shear instability. Localization does not necessarily lead to unstable slip, whereas intergranular dilatation facilitating granular flow does (see the Spiers model, lines 202-218 and Fig. 5 of the revised ms).

(ii) The bottom of the upper crustal seismogenic zones capable of spontaneously nucleating unstable frictional slip is typically associated with temperatures of 250-400 C. If the authors are aware of many (any?) major earthquakes nucleating in regions with the temperature of 550 C, they should cite these examples. Furthermore, if rupture nucleates at any depth, it is much more likely to become a microearthquake than a “major” earthquake, simply because there are a lot more small events nucleating at any depths. So even if the process of microscale cavitation discussed in the manuscript could lead by itself to earthquake nucleation, it is unclear why that would be a mechanism for nucleation of major earthquakes specifically, and not any earthquakes more generally. In fact, the transition from stable to unstable behavior described in the manuscript would be more relevant to observations of microseismic aftershocks after major events that occur deeper than interseismic microseismicity, presumably because major events increase creeping rates at depth.

As mentioned above, we infer on the importance of our results to the nucleation of major earthquakes because we focussed on processes controlling stable vs unstable slip at the lower boundary of the seismogenic zone, where the largest earthquakes frequently nucleate. We agree with the reviewer that the lower limit of the seismogenic zone is usually associated with temperatures of 250-400°C. However, in the present experiments we aimed to reproduce a friction to flow transition in calcite gouge layers (as reported by *Verberne et al., 2015*), to study the deformation processes controlling this transition. The deformation processes in our experiments are likely to be generally relevant to the brittle-ductile transition in faults, including at the lower boundary of the seismogenic zone.

To address the reviewers comment we have clarified our choice of conditions, in particular in the introduction section (lines 35-58 of the revised ms). We explicitly describe how the cavitation mechanism can bring about a shear instability that can lead to runaway slip hence an earthquake (lines 208-218), and we make a note on the expected general applicability of the Spiers model to the upper-crustal seismogenic zone (lines 260-262). We have also changed the title of the ms, to ‘*Major earthquakes triggered by microscale cavitation at the base of the seismogenic zone*’, which we feel better captures the focus of the ms.

2. There are several unclear points about the experimental results and their interpretation.

(i) Why is the temperature of 550 C used? As explained above, it seems to be too high to be

the most relevant, so it is unclear why this and only this temperature is considered. It would help if the manuscript contained some discussion of the thermal structure of the relevant regions, why the temperature of 550 C was chosen, and why other temperatures are not considered. It would be even more illuminating to see some experiments at lower temperatures.

See our reply to previous comments, on the choice of experimental conditions. We addressed the reviewers comment by rewriting the introduction section, specifically clarifying our rationale for the experiments and choice of conditions.

(ii) How are the “steady-state” shear stress values in Figure 1c are determined and what is their uncertainty/error? For some cases, like the effective stresses of 90 and 100 MPa in Figure 1b, stick-slip occurs, so there is no steady-state sliding and hence the “steady-state” shear stress is not obvious. For the case of 50 MPa effective stress in Figure 1b, the shear stress seems to take multiple values, between 25 and 30 MPa in the beginning of the experiment (at about 5 mm slip) and not more than 15 MPa by the end of the experiment (judging by the results for the closely related 60 and 40 MPa for 10-12 mm slips), clearly indicating that the process is quite complex and path-dependent, and there is no single steady-state shear stress value for the effective stress of 50 MPa and slip rate of 10 microns/s. Yet Figure 1c contains unambiguous steady-state shear stress values for all these case; for example, a value slightly higher than 30 MPa is used for the case of 50 MPa effective stress, which is not at all supported by the values from Figure 1b.

The data in revised fig 1c are steady state shear strength values, reached after each (upward) step in σ_n^{eff} -stepping experiments. Because we only used steady state values, not all data could be used, specifically from runs at $v = 10 \mu\text{m/s}$, which frequently showed stick-slip. The error on the absolute strength data is small, i.e. less then 0.1 % on the absolute strength data (see also *Den Hartog et al., 2012* and *Verberne et al., 2015*; we make a note of the size of the error in lines 287-288 (methods).

We thank the reviewer for pointing us to the rather confusing way in which we reported the σ_n^{eff} -stepping data in the original ms. To address this, and other comments made by the reviewer, we now only report data from upward steps in effective normal stress. The downward-stepping data was incomplete (i.e. only present for experiments at $v \geq 1 \mu\text{m/s}$), and frequently showed stick-slips so that we didn't use these shear stress data in the rest of the ms. By showing upward steps only, we declutter the data set, without losing essential information. We present shear stress vs displacement plots of σ_n^{eff} stepping runs at 0.1, 1 and 10 $\mu\text{m/s}$, showing a visually striking difference between runs in the flow regime (0.1 $\mu\text{m/s}$) versus those in the frictional regime (1 and 10 $\mu\text{m/s}$). Plots not shown in Fig. 1 are given in Supplementary Fig. 1. We further improved clarity of the shear stress vs displacement curves (revised Fig. 1) by referring to experiment names in a legend, and in the caption, as listed in Supplementary Table 1. A list of strength data is given in (revised) Supplementary Table 2, where in grey we indicate strength values in the case of stick-slip (thus not plotted in revised Fig. 1c).

(iii) As slip accumulates in the experiment of Figure 1b, the fault gets progressively weaker, as evident from the discussion of the 50 MPa effective stress above. Why is that? Is this the effect of local shear heating and perhaps pore fluid pressurization?

Frictional heating and pore fluid pressurization are not likely to have played a major role in our experiments, because the displacement rates used are very small, at least when compared with the experiments for which these processes are reported (*Di Toro et al., 2011*). Indeed, the displacement rate must have been much higher during an individual slip event in the case of stick-slip, but still, frictional heat or associated pressurized pore fluid is expected to dissipate rapidly during the ‘stick’ phase. Moreover, the weakening we observe in our experiments occurs over several mm’s of displacement, often regardless of the presence of frictional instability (cf. *Den Hartog et al., 2012; Verberne et al., 2015*). We suspect this may be related to gouge layer extrusion from the ring-shaped sample holder during shear.

In the revised ms we chose to limit the effective normal stress stepping data to upward steps only (see our reply to point ii above). The total displacement reached in each experiment now ranges within 8 to 10 mm (revised Fig. 1), and strong weakening trends are absent.

(iv) The results for the slip rate of 0.1 micrometer/s in Figure 1c are described as “ductile flow” “insensitive” to the effective stress (e.g., lines 108-109). Yet the shear resistance goes up significantly with the effective stress in Figure 1c, just not as significantly as for higher slip rates. What is this due to? Does this mean that the behavior for slip rates of 0.1 micrometer/s is a mixture of ductile flow and frictional sliding, not just ductile flow as stated in the manuscript?

Point taken, yes, we agree with the reviewer that the behaviour for slip rates of 0.1 $\mu\text{m/s}$ is essentially a mixture of flow and friction. As mentioned in a reply to a comment made by Reviewer #1, the transition from pure friction to pure ductile flow in sheared gouge is characterized by mixed-mode or ‘frictional-viscous’ flow behaviour (e.g. *Bos & Spiers, 2002; Holdsworth et al., 2001; Imber et al., 2008*). Some dependence of shear strength on effective normal stress is therefore expected, especially close to the transition from (pure) frictional slip as is the case here. To see how we have addressed this point in the revised ms, please refer to our reply to comments made by Reviewer #1 on the same subject.

(v) Is dilatancy measured in the experiments? If yes, it would be good to report it. If not, why not?

We measured layer thickness in some experiments, however these data are frequently incomplete because the sensor measuring layer thickness went out-of-range (which we did not correct for). To measure the onset of dilatation in the flow-to-friction transition, one must enable self-acceleration of the gouge layer, which is technically challenging.

3. Minor comments.

(i) Line 34 and similar occurrences: “velocity weakening” is an adjective quantifying frictional behavior, and hence could use a dash: “velocity-weakening.” In contrast, no dash is fine in line 41, where weakening is a noun.

OK, thanks for pointing this out. In the revised MS, we chose to abbreviate to v-weakening and v-strengthening, to avoid overusing the word ‘velocity’.

(ii) A number of people are thanked in the acknowledgements, but it is not specified for what. OK, thanks. We have rewritten the acknowledgements following the model of a recently published Nature Communications paper.

References cited

- Bos, B., and Spiers, C. J. Frictional-viscous flow of phyllosilicate-bearing fault rock: Microphysical model and implications for crustal strength profiles. *J. Geophys. Res.* **107**, B2, 2028, doi:10.1029/2001JB000301 (2002).
- Chen, J., Verberne, B. A., and Spiers, C. J. Interseismic re-strengthening and stabilization of carbonate faults by “non-Dieterich-type” healing under hydrothermal conditions. *Earth Planet. Sci. Lett.* **423**, 1-12 (2015).
- De Bresser, J. H. P., Evans, B., and Renner, J. (2002). On estimating the strength of calcite rocks under natural conditions. *In: Deformation mechanisms, Rheology and Tectonics: Current status and future perspectives* (eds. De Meer, S., Drury, M. R., De Bresser, J. H. P., and Pennock, G. M.). Geological Society, London, Spec. Pub. **200**, 309-329.
- Den Hartog, S. A. M., Peach, C. J., de Winter, D. A. M., Spiers, C. J. & Shimamoto, T. Frictional properties of megathrust fault gouges at low sliding velocities: New data on effects of normal stress and temperature. *J. Struct. Geol.* **38**, 156-171 (2012).
- Di Toro, G. *et al.* Fault lubrication during earthquakes. *Nature* **471**, 494-498, doi:10.1038/nature09838 (2011).
- Fliervoet, T. F., White, S. H., and Drury, M. R. Evidence for dominant grain boundary sliding deformation in greenschist- and amphibolites grade polymineralic ultramylonites from the Redbank Deformed Zone, Central Australia. *J. Struct. Geol.* **19**, 1495-1520 (1997).
- Fusseis, F., Regenauer-Lieb, K., Liu, J., Hough, R. M., and De Carlo, F. Creep cavitation can establish a dynamic granular fluid pump in ductile shear zones. *Nature* **459**, 974-977 (2009).
- Handy, M. R., Wissing, S. B., and Streit, L. E. Frictional-viscous flow in mylonite with varied biminerale composition and its effect on lithospheric strength. *Tectonophysics* **303**, 175-191 (1999).
- Herwegh, M., Xiao, X. & Evans, B. The effect of dissolved magnesium on diffusion creep in calcite. *Earth Planet. Sci. Lett.* **212**, 457-470 (2003).
- Holdsworth, R. E., Stewart, M., Imber, J. & Strachan, R. A. The structure and rheological evolution of reactivated continental shear zones: a review and case study. *Geol. Soc. London Spec. Publ.* **184**, 115-137 (2001).
- Imber, J., Holdsworth, R. E., Smith, S. A. F., Jefferies, S. P., and Colletini, C. Frictional-viscous flow, seismicity and the geology of weak faults: a review and future directions. *In: The internal structure of fault zones: Implications for mechanical and fluid-flow properties* (eds. Wibberley, C. A. J., Kurz, W., Imber, J., Holdsworth, R., and Colletini, C.). Geological Society, London, Spec. Publ. **299**, 151-173, doi:10.1144/SP299.10 (2008).
- Jin, D. & Karato, S.-I. Mechanisms of shear localization in the continental lithosphere: inference from the deformation microstructures of peridotites from the Ivrea zone, northwestern Italy. *J. Struct. Geol.* **20**, 195-209 (1998).
- Kawamoto, E., and Shimamoto, T. Mechanical behavior of halite and calcite shear zones from brittle to fully-plastic deformation and a revised fault model. *In: Proceedings of the 30th international geological congress, Beijing, China, 4-14 August 1996*, **14**, 89-105, VSP, Utrecht, The Netherlands (1997).
- Kawamoto, E., and Shimamoto, T. The strength profile for biminerale shear zones: an insight from high-temperature shearing experiments on calcite-halite mixtures. *Tectonophysics* **295**, 1-14 (1998).
- Menegon, L., Fusseis, F., Stünitz, H., and Xiao, X. Creep cavitation bands control porosity and fluid flow in lower crustal shear zones. *Geology* **43**, 227-230 (2015).
- Mitchell, E. K., Fialko, Y., and Brown, K. M. Temperature dependence of frictional healing of Westerly granite: Experimental observations and numerical simulations. *Geochem. Geophys. Geosys.* **14**, doi:10.1029/2012GC004241 (2013).
- Morrow, C. A., Radney, B., and Byerlee, J. D. (1992). Frictional strength and the effective pressure law of montmorillonite and illite clays. *In: Fault mechanics and transport properties of rocks* (eds. Evans, B., and Wong, T. F.). Academic Press, pp. 33–67 (London, UK).
- Rempe, M., Smith, S., Mitchell, T., Hirose, T., and Di Toro, G. The effect of water on strain localization in calcite fault gouge sheared at seismic slip rates. *J. Struct. Geol.* **97**, 104-117 (2017).
- Rybacki, E., Wirth, R., and Dresen, G. High-strain creep of feldspar rocks; implications for cavitation and ductile failure in the lower crust. *Geophys. Res. Lett.* **35**, L04304, doi:10.1029/1007GL032478 (2008).

- Shigematsu, N., Fujimoto, K., Ohtani, T., and Goto, K. Ductile fracture of fine-grained plagioclase in the brittle-plastic transition regime: implication for earthquake source nucleation. *Earth Planet. Sci. Lett.* **222**, 1007-1022 (2004).
- Shimamoto, T. Transition between frictional slip and ductile flow for halite shear zones at room temperature. *Science* **231**, 4739, 711-714 (1986).
- Smith, S. A. F., Nielsen, S. and Di Toro, G. Strain localization and the onset of dynamic weakening in calcite fault gouge. *Earth Planet. Sci. Lett.* **413**, 25-36 (2015).
- Takahashi, M., van den Ende, M. P. A., Niemeijer, A. R., and Spiers, C. J. Shear localization in a mature mylonitic rock analog during fast slip. *Geochem. Geophys. Geosys.* **18**, 513-530 (2017).
- Verberne, B. A. *et al.* Frictional properties and microstructure of calcite-rich fault gouges sheared at sub-seismic sliding velocities. *Pure Appl. Geophys.* **171**, 2617-2640 (2013).
- Verberne, B. A., Niemeijer, A. R., De Bresser, J. H. P. & Spiers, C. J. Mechanical behavior and microstructure of simulated calcite fault gouge sheared at 20-600°C: Implications for natural faults in limestones. *J. Geophys. Res.* **120**, 8169-8196 (2015).
- Wintsch, R. P., Christofferson, R., and Kronenberg, A. K. Fluid-rock reaction weakening of fault zones. *J. Geophys. Res.* **100**, B7, 13021-13032 (1995).

REVIEWERS' COMMENTS:

Reviewer #1 (Remarks to the Author):

I (reviewer 1) have reviewed the revised manuscript and associated materials, as well as the response of the authors to comments of both reviewers. Overall, the authors have responded well in rebuttal and in revising the manuscript. Also, the addition of a couple new experiments to demonstrate that the sign of rate dependence changes across the transition from flow to friction strengthens the interpretation appreciably. The paper is now very clearly written and organized, and the interpretations are well supported and argued. I support publication of this paper and believe it will be a solid contribution to developing an improved understanding of processes influencing instability on faults at the base of the seismogenic zone.

In reading the revised manuscript, and the comments of reviewer 2, I have a couple minor comments that may be useful to the authors.

I would like to make two comments on the wording in the abstract, in lines 17-20. The statement "With increasing shear rate or decreasing temperature, dislocation and diffusion creep become too slow to accommodate the imposed shear strain rate, leading to intergranular cavitation, hence weakening, strain localization, and a switch from stable flow to runaway fault rupture." Basically, I don't believe that weakening and strain localization necessarily occurs with a switch from flow by "pure dislocation and diffusion creep" to flow by "dislocation and diffusion creep plus cavitation," though I agree that you have shown this to be the case in your experiments. So in the abstract statement, using "hence," seems wrong in that you have not proved this is a general consequence. I would suggest rewording that a bit.

More importantly, however, is that the statement "with increasing shear rate or decreasing temperature" is not really amplified in the paper, particularly with respect to a concern repeatedly expressed by Reviewer 2. Reviewer 2 rightfully questions the significance of your results to natural faulting in that the transition in rate dependence at 550 C and slip rates of 0.1 micrometer/s, when both these values are too high to compare well with development of instability in natural crustal faulting. In your response and in revisions of the paper, you neglected the opportunity to point out that with trade-offs between temperature and time, it is quite likely that the transition you demonstrate at 550 C likely occurs at lower slip rates at lower temperatures, and as such could very well be consistent with natural conditions at the base of the seismogenic zone for crustal faults in carbonates. You might want to point this out somewhere in the discussion of the paper.

Reviewer #2 (Remarks to the Author):

In the revision, the authors significantly improved the presentation and explanation of their results and prior work on the subject. In my opinion, the manuscript can be published in Nature Communications, with one important change.

As I pointed out in the original review, the title of the original manuscript, "Microscale cavitation as a mechanism for the nucleation of major earthquakes," was disconnected from the findings, as there is nothing in the work that suggests that the described mechanisms of microscale cavitation would in fact nucleate "major" earthquakes.

The new title makes the disconnect even stronger by stating " Major earthquakes triggered by microscale cavitation at the base of the seismogenic zone."

To properly reflect the findings, the title should be something along the lines of "Microscale cavitation as a mechanism for flow-to-friction transition at the base of the seismogenic zone" or at least

"Microscale cavitation as a mechanism for nucleating earthquakes at the base of the seismogenic zone," without the word "major."

Reply to the 2nd Round of Reviewers' Comments

Original submission date: 13 March 2017

Revised MS submission date: 24 August 2017

Manuscript #: NCOMMS-17-06190A

We thank the editor and reviewers for their assessment of our revised manuscript (ms) entitled '*Microscale cavitation as a mechanism for the nucleation of major earthquakes*'. Based on constructive comments made by the reviewers we have revised the manuscript for a second time, and we have edited the manuscript to comply with format requirements for publication as a *Nature Communications Article*. Below we provide point-by-point reply to the reviewers comments of our revised ms.

Reviewers' comments

Reviewer #1 (Remarks to the Author):

I (reviewer 1) have reviewed the revised manuscript and associated materials, as well as the response of the authors to comments of both reviewers. Overall, the authors have responded well in rebuttal and in revising the manuscript. Also, the addition of a couple new experiments to demonstrate that the sign of rate dependence changes across the transition from flow to friction strengthens the interpretation appreciably. The paper is now very clearly written and organized, and the interpretations are well supported and argued. I support publication of this paper and believe it will be a solid contribution to developing an improved understanding of processes influencing instability on faults at the base of the seismogenic zone.

We thank the reviewer for his/ her constructive assessment.

In reading the revised manuscript, and the comments of reviewer 2, I have a couple minor comments that may be useful to the authors.

I would like to make two comments on the wording in the abstract, in lines 17-20. The statement "With increasing shear rate or decreasing temperature, dislocation and diffusion creep become too slow to accommodate the imposed shear strain rate, leading to intergranular cavitation, hence weakening, strain localization, and a switch from stable flow to runaway fault rupture." Basically, I don't believe that weakening and strain localization necessarily occurs with a switch from flow by "pure dislocation and diffusion creep" to flow by "dislocation and diffusion creep plus cavitation," though I agree that you have shown this to be the case in your experiments. So in the abstract statement, using "hence," seems wrong in that you have not proved this is a general consequence. I would suggest rewording that a bit.

OK, point taken. We deleted the word "hence" before "weakening" to avoid over-speculation.

More importantly, however, is that the statement "with increasing shear rate or decreasing temperature" is not really amplified in the paper, particularly with respect to a concern repeatedly expressed by Reviewer 2. Reviewer 2 rightfully questions the significance of your results to natural faulting in that the transition in rate dependence at 550 C and slip rates of 0.1 micrometer/s, when both these values

are too high to compare well with development of instability in natural crustal faulting. In your response and in revisions of the paper, you neglected the opportunity to point out that with trade-offs between temperature and time, it is quite likely that the transition you demonstrate at 550 C likely occurs at lower slip rates at lower temperatures, and as such could very well be consistent with natural conditions at the base of the seismogenic zone for crustal faults in carbonates. You might want to point this out somewhere in the discussion of the paper.

We thank the reviewer for pointing this out. In addition to temperature and time or slip velocity, the effective normal stress is also important in controlling the flow-to-friction transition in gouge-filled faults. We implemented the reviewers' comment by adding a sentence to the fourth paragraph of the Discussion, specifically addressing the importance of trade-offs between temperature, slip velocity, and effective normal stress. We refer to papers by Chester & Higgs (1992) and Den Hartog & Spiers (2013), who discuss these trade-offs in more detail.

Reviewer #2 (Remarks to the Author):

In the revision, the authors significantly improved the presentation and explanation of their results and prior work on the subject. In my opinion, the manuscript can be published in Nature Communications, with one important change.

We thank the reviewer for his/ her constructive assessments, which, as he/ she recognizes, have helped to significantly improve the manuscript.

As I pointed out in the original review, the title of the original manuscript, "Microscale cavitation as a mechanism for the nucleation of major earthquakes," was disconnected from the findings, as there is nothing in the work that suggests that the described mechanisms of microscale cavitation would in fact nucleate "major" earthquakes. The new title makes the disconnect even stronger by stating "Major earthquakes triggered by microscale cavitation at the base of the seismogenic zone."

To properly reflect the findings, the title should be something along the lines of "Microscale cavitation as a mechanism for flow-to-friction transition at the base of the seismogenic zone" or at least "Microscale cavitation as a mechanism for nucleating earthquakes at the base of the seismogenic zone," without the word "major."

We agree with the reviewer, and we have changed the title accordingly. We thank the reviewer for his/ her detailed suggestions.

References

Chester, F. M. & Higgs, N. G. Multimechanism friction constitutive model for ultrafine quartz gouge at hypocentral conditions. *J. Geophys. Res.* **97**, 1859-1870 (1992).

Den Hartog, S. A. M. & Spiers, C. J. Influence of subduction zone conditions and gouge composition on frictional slip stability of megathrust faults. *Tectonophysics* **600**, 75-90 (2013).

Reply to Reviewers' Comments

Original submission date: 13 March 2017

Manuscript #: NCOMMS-17-06190-T

We thank the editor and reviewers for their assessment of our manuscript (ms) entitled '*Microscale cavitation as a mechanism for the nucleation of major earthquakes*'. We have revised the ms on the basis of the constructive comments made by the reviewers, including additional experimental and microstructural data, and rewriting/ reorganizing the main text, figures, as well as the supplement. The revised ms has lengthened, however it has improved significantly in clarity and content, while remaining well within the guidelines for a Nature Communications *Article*. Below we specify how we have improved the ms, addressing each of the points made by the reviewers.

Reviewers' comments

Reviewer #1 (Remarks to the Author):

The topic area and the findings and conclusions have the potential to be significant and timely. The key findings and conclusions of the manuscript are clearly summarized by the statements in the abstract, specifically “Here we show that the flow-to-friction transition in experimentally simulated calcite faults is characterized by a switch from non-dilatant, dislocation and diffusion creep to dilatant deformation, involving incompletely accommodated grain boundary sliding,” and also the statement applying the results to earthquake faulting, specifically, “The observed shear instability, triggered by the onset of microscale cavitation, provides a key mechanism for bringing about the brittle-ductile transition and nucleating (major) earthquakes at depths normally associated with ductile flow.” Unfortunately, after reading the manuscript I do not think the authors have made a convincing case to warrant such statements. I do not think the data and analyses support the conclusions presented in the paper.

We thank the reviewer for recognizing the potential impact of our findings and conclusions. In the revised ms we have addressed the reviewers concerns by providing additional experimental data, as well as by rewriting parts of the main text (see our replies below).

The main observations based on laboratory experiments of shearing granular calcite at high pressure and temperature appropriate to earthquake faulting is the documentation of two distinct modes of shearing behavior, stable shearing at low velocity and unstable (stick-slip) shearing at higher velocity, and that these modes correlate with distributed deformation and very localized deformation, respectively. However, the experimental data summarized in Figure 1c clearly and unequivocally shows that for all the conditions tested the behavior is frictional. That is, for both modes of behavior, the shear strength increases with increase in effective normal stress, and the dependence is quite pronounced consistent with friction.

Moreover, the stick slip behavior showing increase in magnitude of stick-slip stress drops with increasing effective normal stress is a classic characteristic of frictional instability.

We agree with the reviewer that for all experiments summarized in (revised) Fig. 1c, the steady-state shear stress (τ_{ss}) increases with increasing effective normal stress (σ_n^{eff}), suggestive of frictional behaviour. However, the curve to the data from runs at $v = 0.1 \mu\text{m/s}$ clearly flattens-off towards higher effective normal stresses, while for $v \geq 1 \mu\text{m/s}$ we find a positive, near-perfectly linear correlation. This difference in slope of linear versus flattening-off in experiments at relatively high versus low displacement rates strongly suggests a transition with decreasing strain rate from a friction-dominated regime to a flow-dominated regime (cf. e.g. *Shimamoto, 1986; Kawamoto & Shimamoto, 1997, 1998*).

To address the reviewers comment, in the revised ms we have clarified the presentation and interpretation of the effective normal stress stepping data, including references to classic works showing a similar friction-to-flow transition in halite (see e.g. lines 143-155 of the revised ms). Below we provide a detailed list of changes made to the ms, addressing this comment as well as other comments made by the reviewer.

As for the comment on stick-slip behaviour as a classical characteristic of frictional instability, we fully agree. Since we argue that (potentially unstable) velocity weakening frictional behaviour occurs in runs using $v \geq 1 \mu\text{m/s}$, but stable velocity strengthening flow in runs using $v \leq 0.1 \mu\text{m/s}$, our interpretations are in fact fully consistent with the reviewers point. Indeed, frictional instability or stick-slip only occurred in tests using $v \geq 1 \mu\text{m/s}$ (revised Fig. 1 and Supplementary Table 2).

Microstructures in both regimes are consistent with dilatant, frictional behavior.

Microstructures documented for both the distributed-stable shear regime and the localized-unstable shear regime show granular textures with intergranular porosity, and in fact, the images show what appears to be a rather large absolute porosity in both regimes (Figure 2, and Figure 3 in the extended data). Such textures and frictional mechanical behavior is associated with dilatancy, and there is no proof for absence of dilatancy.

OK, point taken. We agree with the reviewer that all recovered microstructures show granular textures with intergranular porosity. However, crucially, in experiments in the frictional regime slip is localized in a narrow band, while in the flow regime, slip is distributed over the entire sample layer. The high shear strain rates applying to a porous, granular shear band (up to 3 s^{-1} in our experiments) strongly suggests dilatant granular flow played a role, consistent with frictional sliding (cf. e.g. *Verberne et al., 2013, 2015; Smith et al., 2015; Rempe et al., 2017*). However, in the flow regime, the much lower shear strain rates ($\sim 10^{-4} \text{ s}^{-1}$) combined with a microstructure composed of porphyroclasts dispersed in a matrix of polygonal grains, is consistent with non-dilatant, creep-controlled flow. This is supported by a comparison with the deformation map of calcite.

To address the reviewers comment, we have clarified our description and interpretation of the mechanical and microstructural data, using explicit references supporting our case. Below we provide a list of revisions specifically addressing this and other points made by the reviewer.

Thus, the first statement quoted from the abstract above, that the distributed-stable shearing regime reflects non-dilatant, dislocation and diffusion creep, is not substantiated. The authors also state, in lines 108-110, that “These mechanical and microstructural observations point to a transition from σ_{eff} -insensitive, ductile flow at low shear rates ($\leq 0.1 \mu\text{m/s}$) to linearly σ_{eff} -dependent, frictional sliding at high shear rates ($\geq 1 \mu\text{m/s}$).” I think the evidence marshaled for inferring diffusion and dislocation processes are important is correct, but not about non-dilatant and non-frictional at low velocities. Also, given that both regimes display intergranular porosity, the second statement quoted from the abstract above, that shear instability is triggered by the onset of cavitation, is also unfounded because cavitation (evidenced by porosity) is present in both regimes.

It is important to emphasize that the transition from pure friction to pure ductile flow in sheared gouge is characterized by mixed-mode or ‘frictional-viscous’ flow behaviour (e.g. *Handy et al., 1999; Bos & Spiers, 2002; Holdsworth et al., 2001; Imber et al., 2008*). Therefore, close to the transition, as is the case in our experiments, a frictional ‘component’ of shear is to be expected. While the transition from pure ductile to friction-viscous flow is gradual, the switch to fully frictional slip, characterized by a near-perfectly linear dependence of shear stress on effective normal stress and velocity weakening behaviour, is abrupt (see also Fig. 6 in the revised ms). This switch occurs because intergranular dilatation or grain-scale cavitation facilitates localization and granular flow, triggering a positive feedback mechanism. Any intergranular dilatation in the frictional-viscous flow regime occurs in insignificant amounts, such that time-dependent deformation processes are fast enough to (sufficiently) compensate and prevent localization or the triggering of an instability.

Multiple lines of evidence are in support of our interpretation of non-dilatant, creep-controlled flow at low rates vs dilatant, frictional slip at high rates in our experiments;

1. Microstructures. Samples sheared at low rates in our experiments show evidence for distributed shear, with porphyroclasts in a matrix of polygonal grains (revised Fig. 2). This is consistent with non-dilatant, creep-controlled flow involving (dynamic) recrystallization (*Fliervoet et al., 1997*). By contrast, at high slip rates we found localized slip in a granular shear band (revised Fig. 3), strongly suggesting that (dilatant) granular flow controlled shear strain accommodation (*Verberne et al., 2013, 2015; Smith et al., 2015; Rempe et al., 2017*)
2. Effective normal stress stepping data. The progressively decreasing slope seen in plots of steady-state shear stress against effective normal stress for experiments at $0.1 \mu\text{m/s}$ points to σ_n^{eff} -insensitive sliding behaviour, characteristic for frictional-viscous flow
3. Calcite deformation maps. Comparison of grain size determined from recovered microstructures and estimates of equivalent compressive stress and strain rate based on our experimental data, with deformation maps for calcite (revised fig 4), strongly suggest that dislocation/ diffusion creep played a role in the low velocity (‘flow regime’) experiments. However, in the high velocity experiments, the shear strain rates applicable to shear band slip are too high.
4. Velocity stepping data. Experiment VS-slow, using $v \leq 0.1 \mu\text{m/s}$, shows intrinsically stable velocity strengthening behaviour, as expected for creep-controlled flow, whereas experiment VS-fast, using $v \geq 1 \mu\text{m/s}$, shows velocity weakening implying frictional sliding. Furthermore, the data from experiment VS-slow demonstrates the gradual nature

of the transition from frictional-viscous to fully ductile flow. The stress exponent or “ n -value”, calculated our v -stepping data (Supplementary Discussion of the revised ms), shows progressively decreasing values from $n = 87$ at $v > 0.03 \mu\text{m/s}$, to $n = 2$ to 4 at $v = 0.001$ to $0.003 \mu\text{m/s}$. High n -values are typically observed in gouge slip experiments conducted in the frictional regime (Mitchell et al., 2013; Chen et al., 2015), whereas n -values of 2 to 4 are characteristic for deformation by dislocation or diffusion creep-controlled flow (De Bresser et al., 2002; Herwegh et al., 2003).

Thanks to the reviewers comments we now realize that in the original ms we had not sufficiently described the lines of evidence described above. To improve this, and other points made by the reviewers, we have revised the ms in the following way;

- i. We have performed new experiments. These additional data on the velocity dependence of sliding in the flow vs. frictional regime strengthen our interpretations (point 4 above). See Fig. 1d, Suppl. Table 1 and 3, Suppl. Fig. 4, and lines 79-88, 162-173, 264-267, 274-276 of the revised ms.
- ii. We have rewritten the main text. We now more explicitly describe our interpretations, including on the gradual nature of the transition from pure ductile to frictional-viscous flow, and on the role of dilatancy in controlling the switch to frictional slip. We use references to relevant previous works to support our interpretations where necessary, and provide more background to explain our rationale and approach. See lines 11-23 (abstract), 25-58 (introduction), 143-155, 162-173, 185-189, 230-234 (discussion) of the revised ms.
- iii. We have reorganized presentation of mechanical data. The effective normal stress stepping data now only contain upward steps in effective normal stress. This declutters the data, while there is no essential information lost. We further improved clarity by referring to experiment names within the figure, as listed in Supplementary Table 1, and we more explicitly describe the data in the main text. Lastly, we now provide shear stress vs displacement plots of all experiments performed, either in the main text or in the Supplement. See Fig. 1b,c, Suppl. Fig. 1, and lines 61-88 of the revised ms.
- iv. We have reorganized figures. In addition to changes made to Fig. 1 (see previous point), we have split the microstructural data into one for the slow and one for the fast experiments. One of the advantages of this is that we can now incorporate EBSD data demonstrating the shear band CPO into the main text, which is quite a technological achievement on such small grains, as well as a rather unique observation. Further, in the revised ms we incorporate the calcite deformation map into the main text, because it convincingly shows the importance of creep-controlled flow in our slow experiments (point 3 above). See Figs. 2, 3, 4 of the revised ms.
- v. We added more microstructural evidence to the Supplement. We now use Supplementary Fig. 4 to show lower magnification BSE images of slow versus fast deformed samples. This provides more evidence for the differences between these experiments, but also helps to show depressurization cracks developed in the slow experiments (lines 104-107 of the revised ms).

Much past experimental work on friction has documented stability transitions with changes in sliding rate, temperature, and normal stress, as well as changes in the steady-state rate

dependence (a-b) with normal stress, temperature and velocity. These works commonly show that for rate and state friction, these transitions can result from progressive but differing changes in the magnitude of the a parameter and the b parameter with velocity, temperature or normal stress. Also, such transitions often, though certainly not necessarily, correlate with changes in shear localization where rate weakening and propensity for instability is correlated with shear localization. Noteworthy in this respect is prior work on halite by several workers including some of these authors. Although the results reported in the paper are interesting, they are not particularly remarkable or new.

Our findings show that localization is crucial in controlling slip stability in calcite at high temperatures. We consider the internal consistency of the present microstructural and mechanical data, as well as with the Spiers model and previous works on halite, quite remarkable. Note that previous studies have not explicitly invoked on the deformation mechanisms active during shear of a widespread, monomineralic crustal fault rock. Moreover, since the deformation mechanisms observed to play a role in our experiments are relevant to any crustal fault rock undergoing a ductile-to-brittle transition, our inferences are expected to be generally applicable (as noted in the concluding paragraph).

To address the reviewers comment, we expanded the introduction to better place our work in relevant context. We now explicitly mention relevant previous work on shear mechanisms at the lower boundary of the seismogenic zone, including on ductile failure (*Shigematsu et al., 2004; Rybacki et al., 2008*) and cavitation (*Fusseis et al., 2009; Menegon et al., 2015*), and recent work hinting on the key role of localization to earthquake nucleation at the lower boundary of the seismogenic zone (*Takahashi et al., 2017*). See lines 53-58 and 227-243 of the revised ms.

In terms of the experiments, overall they are well done. I do have some minor concern whether pore fluid pressure is able to be maintained in the deforming calcite layers under all conditions of testing, and even if it is, what the effective stress law is for deformation involving changes in porosity, intergranular sliding and dislocation/diffusion processes.

While the reviewer has a point concerning the effective normal stress state in for that matter in any gouge sliding experiment that possibly involve a low dynamic permeability (e.g. *Morrow et al., 2000; Verberne et al., 2013*), we expect that this is not a major problem in our experiments. Differences in pore pressure will dissipate, or else steady-state is not likely to be achieved. Since all experiments not showing stick-slip showed near-steady state sliding behaviour, we are confident that our shear stress data are trustworthy. Moreover, as mentioned above, multiple independent lines of evidence support our interpretations, which further strengthens our case.

This study begs for some additional experimental work to test effective stress with different combinations of pore fluid pressure and confining pressure, as well as to actually measure rate dependence of friction directly via velocity stepping.

We share the reviewers' curiosity for doing additional experiments, especially on the rate dependence of strength. We thank the reviewer for pointing this out. Indeed, as mentioned above, we have added v -stepping experiments to the revised ms, one using $v \leq 0.1$

$\mu\text{m/s}$ (VS-slow) and one using $v \geq 1 \mu\text{m/s}$ (VS-fast). See Fig. 1d and lines 79-98, 162-173, 264-267 of the revised ms.

On using different combination of pore fluid pressure and confining pressure, we agree that this is also interesting, however there are limits to what one could and should present in a single manuscript. Moreover, variations in confining pressure cannot be easily tested using the Utrecht Ring shear apparatus (see *Den Hartog et al., 2012; Verberne et al., 2015*).

To some extent, for this paper to focus on frictional instability in terms of rate weakening and dilatancy (or not) without directly measuring either, is problematic. The statement in the abstract that onset of microscale cavitation can nucleate major earthquakes could be correct in some cases, but certainly is not supported by the observations in this paper. Accordingly I cannot recommend this paper for publication.

We thank the reviewer for his/ her constructive comments, which we feel have helped to improve the significantly.

Reviewer #2 (Remarks to the Author):

The manuscript presents laboratory experiments that aim to clarify the nature of shear resistance at depths with relatively high temperatures (~ 550 C) that are typically thought to be stable. The experimental results are interesting and novel, and suggest complex behaviors at different slip rates and effective stresses, including transition from distributed shear and stable slip to shear localization and weakening. They would be of interest to a wide range of scientists working on earthquakes. Hence the work is in principle suitable for publication in *Nature Communications*. At the same time, there are a number of unclear points that require major changes to the text and perhaps additional experiments.

We thank the reviewer for recognizing the potential impact of our work, and its suitability for publication in *Nature Communications*.

1. The claim in the title of the manuscript, “Microscale cavitation as a mechanism for nucleation of major earthquakes, “ does not seem to be supported by the work and discussion in its present form. Briefly, (i) the results presented do not suggest that the layers studied can accelerate from locked to seismic slip rates, the process typically called “nucleation” and (ii) even if such nucleation could spontaneously happen, I am not sure why this process would necessarily nucleate “major earthquakes;” I am not aware of any studies that suggest that any large upper-crustal earthquakes have nucleated in the locations with the relatively high (550 C) temperatures considered in the study.

It is well-established that seismogenesis occurs on faults showing velocity weakening behaviour (e.g. *Scholz, 1998*). By studying the microphysical processes controlling the onset of velocity weakening under conditions pertaining to the lower boundary of seismogenic zone, we study key processes relevant to earthquakes nucleating there. Since the strongest earthquakes frequently nucleate close to the lower boundary of the seismogenic zone (lines

25-27 of the revised ms), we suggest our work is relevant to the nucleation of major earthquakes. We chose 550°C to simulate a friction-to-flow transition in sheared calcite layers, as previously observed to occur around 500 to 600°C in calcite gouge layers sheared under similar conditions of effective normal stress and fluid pressure (*Verberne et al., 2015*).

To address the reviewers comment, in the revised ms provide more background on the importance of the velocity dependence of shear strength in controlling fault stability (lines 35-40), and we better explain our rationale in choosing the experimental conditions (lines 44-52). Also, we provide new data on the velocity dependence of strength, consistent with all other results (see also replies to comments made by Reviewer #1).

Here are some more detailed thoughts on these two points.

(i) Before earthquake nucleation at the bottom of the seismogenic zone, the region in question is likely locked, or perhaps creeping with slip rates comparable to the loading plate rates. The loading plate rates are in tens of millimeters per year and correspond to about $10^{(-9)}$ m/s. According to the experiments, slip at slip rates of 0.1 micrometer/s = $10^{(-7)}$ m/s or lower is stable. Then how would a region at the bottom of the seismogenic zone, creeping with plate-like velocities of $10^{(-9)}$ m/s or lower (if it is partially locked) self-accelerate to nucleate dynamic rupture, if such slip velocities are stable? And if another mechanism is required to start the nucleation and bring it to the slip rates of 1-100 microns/s = $10^{(-6)}$ - $10^{(-4)}$ m/s that are shown to be unstable in this work, then why not attribute the nucleation to that other mechanism?

The reviewer raises an interesting point here. We suggest that a region at the bottom of the seismogenic zone creeping at plate-like velocities may self-accelerate to the velocity-weakening regime, through localization, implying that the processes causing localization are crucial in triggering an instability (e.g. recrystallization, reaction-weakening – *Wintsch et al., 1995; Jin & Karato, 1998*). However, the key process for bringing about a shear instability is grain-scale cavitation, since this facilitates granular flow and velocity weakening, triggering the positive-feedback mechanism or shear instability. Localization does not necessarily lead to unstable slip, whereas intergranular dilatation facilitating granular flow does (see the Spiers model, lines 202-218 and Fig. 5 of the revised ms).

(ii) The bottom of the upper crustal seismogenic zones capable of spontaneously nucleating unstable frictional slip is typically associated with temperatures of 250-400 C. If the authors are aware of many (any?) major earthquakes nucleating in regions with the temperature of 550 C, they should cite these examples. Furthermore, if rupture nucleates at any depth, it is much more likely to become a microearthquake than a “major” earthquake, simply because there are a lot more small events nucleating at any depths. So even if the process of microscale cavitation discussed in the manuscript could lead by itself to earthquake nucleation, it is unclear why that would be a mechanism for nucleation of major earthquakes specifically, and not any earthquakes more generally. In fact, the transition from stable to unstable behavior described in the manuscript would be more relevant to observations of microseismic aftershocks after major events that occur deeper than interseismic microseismicity, presumably because major events increase creeping rates at depth.

As mentioned above, we infer on the importance of our results to the nucleation of major earthquakes because we focussed on processes controlling stable vs unstable slip at the

lower boundary of the seismogenic zone, where the largest earthquakes frequently nucleate. We agree with the reviewer that the lower limit of the seismogenic zone is usually associated with temperatures of 250-400°C. However, in the present experiments we aimed to reproduce a friction to flow transition in calcite gouge layers (as reported by *Verberne et al., 2015*), to study the deformation processes controlling this transition. The deformation processes in our experiments are likely to be generally relevant to the brittle-ductile transition in faults, including at the lower boundary of the seismogenic zone.

To address the reviewers comment we have clarified our choice of conditions, in particular in the introduction section (lines 35-58 of the revised ms). We explicitly describe how the cavitation mechanism can bring about a shear instability that can lead to runaway slip hence an earthquake (lines 208-218), and we make a note on the expected general applicability of the Spiers model to the upper-crustal seismogenic zone (lines 260-262). We have also changed the title of the ms, to '*Major earthquakes triggered by microscale cavitation at the base of the seismogenic zone*', which we feel better captures the focus of the ms.

2. There are several unclear points about the experimental results and their interpretation.

(i) Why is the temperature of 550 C used? As explained above, it seems to be too high to be the most relevant, so it is unclear why this and only this temperature is considered. It would help if the manuscript contained some discussion of the thermal structure of the relevant regions, why the temperature of 550 C was chosen, and why other temperatures are not considered. It would be even more illuminating to see some experiments at lower temperatures.

See our reply to previous comments, on the choice of experimental conditions. We addressed the reviewers comment by rewriting the introduction section, specifically clarifying our rationale for the experiments and choice of conditions.

(ii) How are the “steady-state” shear stress values in Figure 1c are determined and what is their uncertainty/error? For some cases, like the effective stresses of 90 and 100 MPa in Figure 1b, stick-slip occurs, so there is no steady-state sliding and hence the “steady-state” shear stress is not obvious. For the case of 50 MPa effective stress in Figure 1b, the shear stress seems to take multiple values, between 25 and 30 MPa in the beginning of the experiment (at about 5 mm slip) and not more than 15 MPa by the end of the experiment (judging by the results for the closely related 60 and 40 MPa for 10-12 mm slips), clearly indicating that the process is quite complex and path-dependent, and there is no single steady-state shear stress value for the effective stress of 50 MPa and slip rate of 10 microns/s. Yet Figure 1c contains unambiguous steady-state shear stress values for all these case; for example, a value slightly higher than 30 MPa is used for the case of 50 MPa effective stress, which is not at all supported by the values from Figure 1b.

The data in revised fig 1c are steady state shear strength values, reached after each (upward) step in σ_n^{eff} -stepping experiments. Because we only used steady state values, not all data could be used, specifically from runs at $v = 10 \mu\text{m/s}$, which frequently showed stick-slip. The error on the absolute strength data is small, i.e. less than 0.1 % on the absolute strength data

(see also *Den Hartog et al., 2012* and *Verberne et al., 2015*; we make a note of the size of the error in lines 287-288 (methods).

We thank the reviewer for pointing us to the rather confusing way in which we reported the σ_n^{eff} -stepping data in the original ms. To address this, and other comments made by the reviewer, we now only report data from upward steps in effective normal stress. The downward-stepping data was incomplete (i.e. only present for experiments at $v \geq 1 \mu\text{m/s}$), and frequently showed stick-slips so that we didn't use these shear stress data in the rest of the ms. By showing upward steps only, we declutter the data set, without losing essential information. We present shear stress vs displacement plots of σ_n^{eff} stepping runs at 0.1, 1 and 10 $\mu\text{m/s}$, showing a visually striking difference between runs in the flow regime (0.1 $\mu\text{m/s}$) versus those in the frictional regime (1 and 10 $\mu\text{m/s}$). Plots not shown in Fig. 1 are given in Supplementary Fig. 1. We further improved clarity of the shear stress vs displacement curves (revised Fig. 1) by referring to experiment names in a legend, and in the caption, as listed in Supplementary Table 1. A list of strength data is given in (revised) Supplementary Table 2, where in grey we indicate strength values in the case of stick-slip (thus not plotted in revised Fig. 1c).

(iii) As slip accumulates in the experiment of Figure 1b, the fault gets progressively weaker, as evident from the discussion of the 50 MPa effective stress above. Why is that? Is this the effect of local shear heating and perhaps pore fluid pressurization?

Frictional heating and pore fluid pressurization are not likely to have played a major role in our experiments, because the displacement rates used are very small, at least when compared with the experiments for which these processes are reported (*Di Toro et al., 2011*). Indeed, the displacement rate must have been much higher during an individual slip event in the case of stick-slip, but still, frictional heat or associated pressurized pore fluid is expected to dissipate rapidly during the 'stick' phase. Moreover, the weakening we observe in our experiments occurs over several mm's of displacement, often regardless of the presence of frictional instability (cf. *Den Hartog et al., 2012*; *Verberne et al., 2015*). We suspect this may be related to gouge layer extrusion from the ring-shaped sample holder during shear.

In the revised ms we chose to limit the effective normal stress stepping data to upward steps only (see our reply to point ii above). The total displacement reached in each experiment now ranges within 8 to 10 mm (revised Fig. 1), and strong weakening trends are absent.

(iv) The results for the slip rate of 0.1 micrometer/s in Figure 1c are described as "ductile flow" "insensitive" to the effective stress (e.g., lines 108-109). Yet the shear resistance goes up significantly with the effective stress in Figure 1c, just not as significantly as for higher slip rates. What is this due to? Does this mean that the behavior for slip rates of 0.1 micrometer/s is a mixture of ductile flow and frictional sliding, not just ductile flow as stated in the manuscript?

Point taken, yes, we agree with the reviewer that the behaviour for slip rates of 0.1 $\mu\text{m/s}$ is essentially a mixture of flow and friction. As mentioned in a reply to a comment made by Reviewer #1, the transition from pure friction to pure ductile flow in sheared gouge is characterized by mixed-mode or 'frictional-viscous' flow behaviour (e.g. *Bos & Spiers, 2002*; *Holdsworth et al., 2001*; *Imber et al., 2008*). Some dependence of shear strength on effective

normal stress is therefore expected, especially close to the transition from (pure) frictional slip as is the case here. To see how we have addressed this point in the revised ms, please refer to our reply to comments made by Reviewer #1 on the same subject.

(v) Is dilatancy measured in the experiments? If yes, it would be good to report it. If not, why not?

We measured layer thickness in some experiments, however these data are frequently incomplete because the sensor measuring layer thickness went out-of-range (which we did not correct for). To measure the onset of dilatation in the flow-to-friction transition, one must enable self-acceleration of the gouge layer, which is technically challenging.

3. Minor comments.

(i) Line 34 and similar occurrences: “velocity weakening” is an adjective quantifying frictional behavior, and hence could use a dash: “velocity-weakening.” In contrast, no dash is fine in line 41, where weakening is a noun.

OK, thanks for pointing this out. In the revised MS, we chose to abbreviate to v-weakening and v-strengthening, to avoid overusing the word ‘velocity’.

(ii) A number of people are thanked in the acknowledgements, but it is not specified for what.

OK, thanks. We have rewritten the acknowledgements following the model of a recently published Nature Communications paper.

References cited

- Bos, B., and Spiers, C. J. Frictional-viscous flow of phyllosilicate-bearing fault rock: Microphysical model and implications for crustal strength profiles. *J. Geophys. Res.* **107**, B2, 2028, doi:10.1029/2001JB000301 (2002).
- Chen, J., Verberne, B. A., and Spiers, C. J. Interseismic re-strengthening and stabilization of carbonate faults by “non-Dieterich-type” healing under hydrothermal conditions. *Earth Planet. Sci. Lett.* **423**, 1-12 (2015).
- De Bresser, J. H. P., Evans, B., and Renner, J. (2002). On estimating the strength of calcite rocks under natural conditions. *In: Deformation mechanisms, Rheology and Tectonics: Current status and future perspectives* (eds. De Meer, S., Drury, M. R., De Bresser, J. H. P., and Pennock, G. M.). Geological Society, London, Spec. Pub. **200**, 309-329.
- Den Hartog, S. A. M., Peach, C. J., de Winter, D. A. M., Spiers, C. J. & Shimamoto, T. Frictional properties of megathrust fault gouges at low sliding velocities: New data on effects of normal stress and temperature. *J. Struct. Geol.* **38**, 156-171 (2012).
- Di Toro, G. *et al.* Fault lubrication during earthquakes. *Nature* **471**, 494-498, doi:10.1038/nature09838 (2011).
- Fliervoet, T. F., White, S. H., and Drury, M. R. Evidence for dominant grain boundary sliding deformation in greenschist- and amphibolites grade polyminerale ultramylonites from the Redbank Deformed Zone, Central Australia. *J. Struct. Geol.* **19**, 1495-1520 (1997).
- Fusseis, F., Regenauer-Lieb, K., Liu, J., Hough, R. M., and De Carlo, F. Creep cavitation can establish a dynamic granular fluid pump in ductile shear zones. *Nature* **459**, 974-977 (2009).
- Handy, M. R., Wissing, S. B., and Streit, L. E. Frictional-viscous flow in mylonite with varied biminerale composition and its effect on lithospheric strength. *Tectonophysics* **303**, 175-191 (1999).
- Herwegh, M., Xiao, X. & Evans, B. The effect of dissolved magnesium on diffusion creep in calcite. *Earth Planet. Sci. Lett.* **212**, 457-470 (2003).
- Holdsworth, R. E., Stewart, M., Imber, J. & Strachan, R. A. The structure and rheological evolution of reactivated continental shear zones: a review and case study. *Geol. Soc. London Spec. Publ.* **184**, 115-137 (2001).
- Imber, J., Holdsworth, R. E., Smith, S. A. F., Jefferies, S. P., and Collettini, C. Frictional-viscous flow, seismicity and the geology of weak faults: a review and future directions. *In: The internal structure of fault zones: Implications for mechanical and fluid-flow properties* (eds. Wibberley, C. A. J., Kurz, W.,

- Imber, J., Holdsworth, R., and Collettini, C.). Geological Society, London, Spec. Publ. **299**, 151-173, doi:10.1144/SP299.10 (2008).
- Jin, D. & Karato, S.-I. Mechanisms of shear localization in the continental lithosphere: inference from the deformation microstructures of peridotites from the Ivrea zone, northwestern Italy. *J. Struct. Geol.* **20**, 195-209 (1998).
- Kawamoto, E., and Shimamoto, T. Mechanical behavior of halite and calcite shear zones from brittle to fully-plastic deformation and a revised fault model. In: Proceedings of the 30th international geological congress, Beijing, China, 4-14 August 1996, **14**, 89-105, VSP, Utrecht, The Netherlands (1997).
- Kawamoto, E., and Shimamoto, T. The strength profile for biminerale shear zones: an insight from high-temperature shearing experiments on calcite-halite mixtures. *Tectonophysics* **295**, 1-14 (1998).
- Menegon, L., Fusses, F., Stünitz, H., and Xiao, X. Creep cavitation bands control porosity and fluid flow in lower crustal shear zones. *Geology* **43**, 227-230 (2015).
- Mitchell, E. K., Fialko, Y., and Brown, K. M. Temperature dependence of frictional healing of Westerly granite: Experimental observations and numerical simulations. *Geochem. Geophys. Geosys.* **14**, doi: 10.1029/2012GC004241 (2013).
- Morrow, C. A., Radney, B., and Byerlee, J. D. (1992). Frictional strength and the effective pressure law of montmorillonite and illite clays. In: Fault mechanics and transport properties of rocks (eds. Evans, B., and Wong, T. F.). Academic Press, pp. 33–67 (London, UK).
- Rempe, M., Smith, S., Mitchell, T., Hirose, T., and Di Toro, G. The effect of water on strain localization in calcite fault gouge sheared at seismic slip rates. *J. Struct. Geol.* **97**, 104-117 (2017).
- Rybacki, E., Wirth, R., and Dresen, G. High-strain creep of feldspar rocks; implications for cavitation and ductile failure in the lower crust. *Geophys. Res. Lett.* **35**, L04304, doi:10.1029/1007GL032478 (2008).
- Shigematsu, N., Fujimoto, K., Ohtani, T., and Goto, K. Ductile fracture of fine-grained plagioclase in the brittle-plastic transition regime: implication for earthquake source nucleation. *Earth Planet. Sci. Lett.* **222**, 1007-1022 (2004).
- Shimamoto, T. Transition between frictional slip and ductile flow for halite shear zones at room temperature. *Science* **231**, 4739, 711-714 (1986).
- Smith, S. A. F., Nielsen, S. and Di Toro, G. Strain localization and the onset of dynamic weakening in calcite fault gouge. *Earth Planet. Sci. Lett.* **413**, 25-36 (2015).
- Takahashi, M., van den Ende, M. P. A., Niemeijer, A. R., and Spiers, C. J. Shear localization in a mature mylonitic rock analog during fast slip. *Geochem. Geophys. Geosys.* **18**, 513-530 (2017).
- Verberne, B. A. *et al.* Frictional properties and microstructure of calcite-rich fault gouges sheared at sub-seismic sliding velocities. *Pure Appl. Geophys.* **171**, 2617-2640 (2013).
- Verberne, B. A., Niemeijer, A. R., De Bresser, J. H. P. & Spiers, C. J. Mechanical behavior and microstructure of simulated calcite fault gouge sheared at 20-600°C: Implications for natural faults in limestones. *J. Geophys. Res.* **120**, 8169-8196 (2015).
- Wintsch, R. P., Christofferson, R., and Kronenberg, A. K. Fluid-rock reaction weakening of fault zones. *J. Geophys. Res.* **100**, B7, 13021-13032 (1995).